# Integrated QTL Mapping, Meta-Analysis, and RNA-Sequencing Reveal Candidate Genes for Maize Deep-Sowing Tolerance

**DOI:** 10.3390/ijms24076770

**Published:** 2023-04-05

**Authors:** Xiaoqiang Zhao, Yining Niu, Zakir Hossain, Jing Shi, Taotao Mao, Xiaodong Bai

**Affiliations:** 1State Key Laboratory of Aridland Crop Science, College of Agronomy, Gansu Agricultural University, Lanzhou 730070, China; 1983shijing@163.com (J.S.); m15294392789@163.com (T.M.); bxd15293898130@163.com (X.B.); 2Swift Current Research and Development Centre, Agriculture and Agri-Food Canada, Swift Current, SK S9H 3X2, Canada; zakir.hossain@agr.gc.ca

**Keywords:** maize, deep-sowing, mesocotyl/coleoptile, quantitative trait loci, meta-analysis, RNA-sequencing, candidate genes, interconnected networks, quantitative real-time PCR

## Abstract

Synergetic elongation of mesocotyl and coleoptile are crucial in governing maize seedlings emergence, especially for the maize sown in deep soil. Studying the genomic regions controlling maize deep-sowing tolerance would aid the development of new varieties that are resistant to harsh conditions, such as drought and low temperature during seed germination. Using 346 F_2:3_ maize population families from W64A × K12 cross at three sowing depths, we identified 33 quantitative trait loci (QTLs) for the emergence rate, mesocotyl, coleoptile, and seedling lengths via composite interval mapping (CIM). These loci explained 2.89% to 14.17% of phenotypic variation in a single environment, while 12 of 13 major QTLs were identified at two or more sowing environments. Among those, four major QTLs in Bin 1.09, Bin 4.08, Bin 6.01, and Bin 7.02 supported pleiotropy for multiple deep-sowing tolerant traits. Meta-analysis identified 17 meta-QTLs (MQTLs) based on 130 original QTLs from present and previous studies. RNA-Sequencing of mesocotyl and coleoptile in both parents (W64A and K12) at 3 cm and 20 cm sowing environments identified 50 candidate genes expressed differentially in all major QTLs and MQTLs regions: six involved in the circadian clock, 27 associated with phytohormones biosynthesis and signal transduction, seven controlled lignin biosynthesis, five regulated cell wall organization formation and stabilization, three were responsible for sucrose and starch metabolism, and two in the antioxidant enzyme system. These genes with highly interconnected networks may form a complex molecular mechanism of maize deep-sowing tolerance. Findings of this study will facilitate the construction of molecular modules for deep-sowing tolerance in maize. The major QTLs and MQTLs identified could be used in marker-assisted breeding to develop elite maize varieties.

## 1. Introduction

Maize (*Zea mays* L.) is one of the most important crops worldwide; it provides human and animal foods and industrial raw materials. In general, maize seedlings are often grown in numerous adverse environments, including those with drought and low temperature during seed germination. Relative rates of yield increase in maize are slowing and are estimated at about 1.5% per year [1]. Meanwhile, climate change has negatively impacted agricultural production (http://faostat.fao.org/; accessed on 16 January 2023), which will not meet the needs of a rapidly increasing global population in the future; the mean growth rate of crop yield must exceed 2.4% per year to feed an estimated global population of 10 billion by the 2050s [2]. To overcome the stagnation or negative trend of maize yield increase due to environmental changes, it is necessary to develop maize varieties that are more resistant to changing environmental conditions. For example, “40107” and “P1213733” maize varieties have a strong deep-sowing tolerance in the deep soil layer (>20 cm), absorbing water from the moist soil, and thus effectively avoiding drought stress at the early seedling stage [3,4,5]. Indeed, the development of maize photosynthetic apparatus remains undisturbed through protection of the shoot apex from cold exposure at a deeper soil level [6]. A deeper soil environment may also allow maize to evade the negative effects of chemical residues in shallow soil as well as damage by wild animals [7], thereby stabilizing the primary form of seedlings [8]. However, due to the inherent characteristics, most existing varieties have weak germination potential in deep soil environments [9]. Thus, improving deep-sowing tolerance would be a key strategy to maintain normal growth and high yield of maize under cold or drought conditions.

Different crop species have evolved diverse organs to adapt to deep-sowing stressors, including elongated mesocotyl in rice (*Oryza sativa* L.) [10] and elongated coleoptile in wheat (*Triticum aestivum* L.) [11] and barley (*Hordeum vulgare* L.) [12]. These structures help the emergence of the shoot tips and the first internode from the embryo to the soil surface; the co-elongated mesocotyl and coleoptile in maize send the buds to the soil surface [13,14], where natural light represses the continued elongation of mesocotyl and coleoptile and induces leaf expansion [7,15]. However, different crops have different degrees of deep-sowing tolerance due to genetic variations. For example, Cornbelt lines in the United States and Canada have longer mesocotyl compared with tropical maize germplasms [16]; the mesocotyl of indica rice is longer than that of japonica rice, and the mesocotyl of weedy rice is longer than that of cultivated rice [17]. These imply that the differences in the deep-sowing tolerance of different crops result from both geographical selection and the breeder’s selection. Therefore, selection of deep-sowing tolerant germplasms for crop improvement is possible.

The source of genetic variation in deep-sowing tolerance has been explored using quantitative trait locus (QTL) mapping in bi-parental maize, mapping populations in a limited number of studies. Studied deep-sowing tolerant traits included emergence rate (RAT), mesocotyl length (MESL), coleoptile length (COLL), plumule length (PL), and seedling length (SDL) at three sowing depths of 12.5 cm, 15 cm, and 20 cm [18,19,20,21]. Although several QTLs controlling these deep-sowing tolerant traits have been identified in maize, there are large genetic distances of these target traits due to the low density of molecular markers and limited population size. Therefore, there has been virtually no progress in the isolation of major QTLs by QTL fine mapping or map-based cloning for marker-assisted selection (MAS) breeding. At the same time, meta-QTL (MQTL) analysis has been used to estimate consensus QTLs linked to genetic factors underlying deep-sowing tolerant traits among multiple independent QTL mapping experiments [22,23,24]. Combined with high-throughput RNA-sequencing (RNA-Seq), MQTLs may quickly identify more effective candidate genes in corresponding regions and elucidate their function and regulatory networks [21]. We previously identified a constitutive-QTL (cQMES4) for MESL in the 160.98–176.22 Mb region of chromosome 4 and validated 15 candidate genes in the region by joint analysis of QTL mapping, bulked-segregant analysis (BSA), and RNA-Seq [21]. Based on the above considerations, the objectives of our study were to: (1) identify QTLs information responsible for six deep-sowing tolerant traits in 346 F_2:3_ population families derived from a W64A × K12 cross at three sowing depths (3 cm, 15 cm, and 20 cm); (2) incorporate previously published data in a meta-analysis to integrate genetic maps and identify corresponding MQTLs involved in deep-sowing tolerance in maize; (3) mine candidate genes in corresponding MQTLs and major QTLs regions by combining RNA-Seq and quantitative real-time PCR (qRT-PCR) in mesocotyl and coleoptile of both parents (W64A and K12) at 3 cm and 20 cm sowing depth environments; and (4) further explore the interconnected networks of candidate genes and interpret the biological processes and pathways controlling maize deep-sowing tolerance. These findings will provide valuable information by revealing the molecular basis of deep-sowing tolerance in maize and annotation of associated genes that will lay a foundation for further functional analysis of genes, thereby regulating deep-sowing tolerance in maize and the discovery of alleles to be used in developing deep-sowing resistant varieties.

## 2. Results

### 2.1. Phenotypic Variations of Six Deep-Sowing Tolerant Traits

We analyzed the differences in RAT, MESL, COLL, SDL, total length of mesocotyl and coleoptile (MESL + COLL), and length ratio of mesocotyl to coleoptile (MESL/COLL) across two parents (female parent, W64A; male parent, K12), an F_1_ hybrid, and an F_2:3_ population at three sowing depths to generate phenotypic data for QTL mapping. Compared with a normal sowing depth of 3 cm, RAT was decreased by a mean of 3.4%, 93.1%, 0.7%, and 12.3% at a deep-sowing stress of 15 cm, and 20.3%, 100.0%, 8.0%, and 38.4% at 20 cm sowing depth in W64A, K12, F_1_ hybrid, and F_2:3_ population, respectively (Table 1). Similarly, SDL was decreased by a mean of 15.3%, 46.3%, 4.3%, and 23.3% at 15 cm, and 35.7%, 63.3%, 28.4%, and 46.5% at 20 cm, respectively (Table 1). Conversely, MESL was increased by a mean of 169.1%, 41.4%, 233.8%, and 192.8% at 15 cm, and 232.5%, 124.1%, 293.8%, and 250.0% at 20 cm, respectively (Table 1). COLL was increased by a mean by 30.9%, 110.9%, 73.4%, and 71.3% at 15 cm, and 55.1%, 120.4%, 93.7%, and 113.0% at 20 cm, respectively (Table 1). MESL + COLL was increased by a mean of 106.8%, 80.4%, 161.7%, and 151.6% at 15 cm, and 152.4%, 122.1%, 204.0%, and 225.3% at 20 cm, respectively (Table 1). In addition, MESL/COLL in W64A, F_1_ hybrid, and F_2:3_ populations, respectively, were increased by a mean of 108.2%, 92.6%, and 11.3% at 15 cm and 114.8%, 103.3%, and 32.3% at 20 cm, while this trait in K12 was decreased by 32.9% at 15 cm and slightly increased (1.3%) at 20 cm (Table 1). These findings demonstrated that deep-sowing was not conducive to maize seedling emergence and subsequent growth, although a certain degree of deep-sowing stress could stimulate mesocotyl and coleoptile elongation in maize.

The values of all six deep-sowing tolerant traits were distributed normally in the F_2:3_ population at all three sowing depths, with absolute values of skewness and kurtosis of <1.0, except for COLL and MESL/COLL at 20 cm depth (skewness = 1.087 and 0.913, kurtosis = 0.934 and 1.017, respectively) (Table 1). The data indicated that the six traits were quantitative in nature and were controlled by multiple QTLs and genes in maize. In addition, the estimated broad-sense heritability ($H_{B}^{2}$) values of the six traits in the F_2:3_ population ranged from 77.6% to 91.6%, and their genotype × environment interaction heritability ($H_{GE}^{2}$) values varied from 13.5% to 25.9%, respectively (Figure 1A), which were consistent with the results from previous studies [5,18,19,20,21]. Our results demonstrated high heritability of these deep-sowing tolerant traits and smaller effects of genotype × environment interaction.

### 2.2. Heterosis Analysis of Six Deep-Sowing Toletant Traits

Heterosis refers to the phenomenon that traits such as growth, yield, fertility, or adverse environmental tolerance in hybrid progeny are improved or increased compared with the parents [25]. Therefore, an in-depth understanding of heterosis in maize deep-sowing tolerant traits could effectively improve the efficiency of crossbreeding for deep-sowing tolerance. We found that the six deep-sowing tolerant traits in maize had different heterosis indices: the F_1_ heterosis index (HI) ranged from 106.0% (COLL) to 177.0% (RAT), the relative heterosis (RH) ranged from 5.7% (COLL) to 36.2% (MESL), the mid-parent heterosis (MH) ranged from 6.0% (COLL) to 77.0% (RAT), and the over-parent heterosis (OH) ranged from −4.3% (MESL/COLL) to 19.9% (COLL) (Figure 1B). These results showed that RAT and MESL had significant MH (>60%), while all traits except MESL/COLL displayed positive OH. In addition, the F_2:3_ advantage reduction rate (ARR) of the six traits decreased by different degrees, and the most was in SDL (−43%) (Figure 1B), which may be closely associated with the F_2_ ARR of plant height in maize that was 21.0% [26].

### 2.3. Framework for Relationships among Six Deep-Sowing Toletant Traits

The examination of phenotypic correlations among six deep-sowing tolerant traits may be useful for interpreting the co-localization of QTLs, which would more likely reveal their genetic relationships. For this, principal component analysis (PCA), Pearson pairwise correlation analysis, and multiple linear regression analysis were performed in two parents, F_1_ hybrid, and F_2:3_ population at different sowing depths conditions. PCA analysis showed that first two principal components (PCs, PC1 and PC2) accounted for 94.1% of the total variance under three environments, of which the eigenvalues were larger than 1.0 (Figure 2A). We therefore speculated that the two PCs linear combinations of different traits were based on their variable loadings. Specifically, MESL, COLL, MESL + COLL, and MESL/COLL were the primary traits in PC1, which accounted for 52.7% of the total variance and represented the developmental characteristics of mesocotyl and coleoptile in maize; RAT and SDL were the primary traits in PC2, which accounted for 41.4% of the total variance and indicated the emergence and growth of maize seedlings (Figure 2B). Pairwise Pearson correlations showed that six deep-sowing tolerant traits produced complementary information at 15 cm and 20 cm deep-sowing environments, and each trait had a significantly positive or negative correlation (*p* < 0.05) with 2–5 other traits (Figure 2C). These data suggested that these traits may have overlapping common genetic bases to regulate maize deep-sowing tolerance. In addition, six deep-sowing tolerant traits of two parents, F_1_ hybrid, and F_2:3_ population at 15 cm and 20 cm sowing depths treatments were standardized, respectively, and then four multiple optimal regression equations were constructed to predict the impact of four mesocotyl and coleoptile growth parameters (MESL, COLL, MESL + COLL, and MESL/COLL) on maize seedling emergence and growth (RAT and SDL), respectively (Figure 2D).

RAT, MESL, COLL, and MESL + COLL of the progeny populations (F_1_ hybrid and F_2:3_ population) depicted a strong positive correlation to both female (W64A) and male parents (K12) (*p* < 0.05) (Table 2), revealed that these four traits in the progeny populations were simultaneously controlled by both parents. The effects of the female parents on MESL, COLL, and MESL + COLL were comparatively larger than the male parents; MESL/COLL had a significantly positive correlation to the female parent (*p* < 0.05), while SDL showed a positive correlation to male parent (*p* < 0.05) (Table 2). In breeding practice, we should therefore comprehensively consider the contributions of both parents to progeny populations and cultivate elite deep-sowing tolerant maize varieties.

### 2.4. QTL Analysis of Six Deep-Sowing Toletant Traits

Previously, we used 253 genome-wide polymorphic simple sequence repeats (SSRs) for genotyping the F_2_ indivuals to construct a genetic linkage map, which spanned a total length of 1410.6 cm with an average interval of 5.6 cm between markers [21]. Subsequently, we detected QTLs using a composite interval mapping (CIM) approach and the threshold logarithm of odds (LOD) score of 3.00 (*p* < 0.05) in Windows QTL Cartographer software v.2.5 (https://brcwebportal.cos.ncsu.edu/qtlcart/WQTLCart.htm; accessed on 21 July 2022). We identified 33 QTLs for six deep-sowing tolerant traits (four for RAT, seven for MESL, five for COLL, eight for MESL + COLL, three for MESL/COLL, and six for SDL) in the F_2:3_ population at three sowing depths, which were distributed over the ten chromosomes; phenotypic variation explained (PVE) within each sowing environment by these QTLs were 2.89% (qMESL–Ch.1–1 at 3 cm environment)—14.17% (qSDL–Ch.7–1 at 20 cm environment) (Figure 3; Appendix A). The QTLs for RAT, MESL, COLL, MESL + COLL, and SDL displayed both additive and non-additive effects, including partial-dominance, dominance, and over-dominance, whereas the QTLs for MESL/COLL only showed non-additive effects (Appendix A). In addition, we detected only one major QTL (qMESL–Ch.1–2) under a single environment (PVE > 10%), and other 12 major QTLs under two or more environments (Appendix A; Figure 3). Of them, qMESL–Ch.1–1 (PVE, 2.89–10.01%), qCOLL–Ch.1–1 (PVE, 10.38–14.12%), and qMESL + COLL–Ch.1–2 (PVE, 5.80–9.37%) were simultaneously located in the Bin 1.09 (umc2047–bnlg1597) region; qMESL/COLL–Ch.4–1 (PVE, 8.43–10.69%), qMESL + COLL–Ch.4–1 (PVE, 4.08–9.40%), and qMESL–Ch.4–2 (PVE, 6.98–8.42%) were simultaneously located in the Bin 4.08 (umc1612–umc1313) region; qMESL–Ch.6–1 (PVE, 5.06–10.03%) and qMESL + COLL–Ch.6–1 (PVE, 5.81–8.15%) were simultaneously located in the Bin 6.01 (umc2311–umc2196) region; qSDL–Ch.7–1 (PVE, 6.26–14.17%); and qMESL–Ch.7–1 (PVE, 3.67–4.93%) were simultaneously located in the Bin 7.02 (umc1585–umc2526) region (Appendix A; Figure 3). These findings confirmed that the four major QTLs regions, i.e., Bin 1.09, Bin 4.08, Bin 6.01, and Bin 7.02 had pleiotropic effects on multiple deep-sowing tolerant traits, including MESL, COLL, MESL + COLL, MESL/COLL, and SDL in maize. 

### 2.5. Consensus Map Development and Meta-QTL Analysis

We established a consensus map using data from three original QTL studies to identify MQTLs for deep-sowing tolerance in maize, and then predicted candidate genes in corresponding MQTLs regions to lay the foundation for fine mapping and candidate gene cloning. In those three QTL studies, the size of the mapping populations ranged from 221 to 346 individuals, the map total length of mapping populations varied from 1410.6 cm to 6242.7 cm, and 130 original QTLs (23 for RAT, 38 for MESL, 15 for COLL, 18 for MESL + COLL, 7 for MESL/COLL, 5 for PL, and 24 for SDL) were identified at five sowing depth environments (Table 3). These QTLs accounted for 22.4%, 4.8%, 9.6%, 12.8%, 7.2%, 10.4%, 8.8%, and 8.0% (variations) on ten chromosomes, respectively (Figure 4A). The LOD scores of each QTL varied from 2.29 to 14.84, with an average of 5.94 (Figure 4B), and the PVE of each QTL ranged from 2.75% to 17.95%, with an average of 7.39% (Figure 4C). We then constructed the consensus linkage map using BioMercator v.4.2.3 (https://urgi.versailles.inra.fr/Tools/BioMercator-V4; accessed on 3 December 2022) with the IBM2 2008 Neighbors Map Frame 6 reference map (https://www.maizegdb.org/data_center/map; accessed on 1 December 2022), which was composed of 1612 markers spanning 7037.79 cm, and 121 original QTLs (approximately 93.1%) were projected on this consensus map (Appendix A).

Using a meta-analysis, we identified 17 MQTLs on ten chromosomes (Table 4; Figure 5); nine detected major QTLs (i.e., qMESL + COLL–Ch.1–1, qMESL–Ch.1–1, qCOLL–Ch.1–1, qSDL–Ch.1–3, qMESL–Ch.1–2, qMESL + COLL–Ch.2–1, qMESL–Ch.3–1, qMESL–Ch.4–1, and qSDL–Ch.7–1) in our F_2:3_ population, which were contained in six MQTLs (i.e., MQTL1–3, MQTL1–4, MQTL2–1, MQTL3–1, MQTL4–1, and MQTL4–1) regions (Table 4; Figure 5), while four other major QTLs (i.e., qSDL–Ch.1–2, qMESL/COLL–Ch.4–1, qMESL–Ch.6–1, and qRAT–Ch.7–1) did not overlap with any MQTLs regions (Table 4; Figure 5). Mining candidate genes in these major QTLs and MQTLs regions might discover novel genes and provide new information on the regulation of deep-sowing tolerance in maize.

### 2.6. Identification of Candidate Genes via RNA-Seq

From the physical distance of markers, we successfully projected 13 major QTLs in our F_2:3_ population and 17 MQTLs to B73 RefGen_v3 physical map (https://www.maizegdb.org/genome/assembly/B73%20RefGen_v3; accessed on 10 December 2022). We then conducted RNA-Seq using 12 mesocotyls and 12 coleoptiles from both W64A and K12 that were cultured for 10 days at 3 cm and 20 cm depths, with three replicates [13,14]. All gene expressions were expressed as the fragments per kilobase of the transcript per million mapped read (FPKM) values, and 60,395 expressed genes were identified (Figure 6A). Based on the FPKM expression differences of the genes in the mesocotyl and coleoptile of both parents in the two environments (Figure 6B) and their functional annotations (Appendix A), we identified 50 candidate genes in above major QTLs and MQTLs regions (Table 4; Figure 5). Identified genes were involved in the circadian clock, multiple phytohormones biosynthesis and signal transduction, lignin biosynthesis, cell wall organization formation and stabilization, sucrose and starch metabolism, and the antioxidant enzyme system (Figure 6B; Appendix A). Moreover, the interaction networks analysis showed that these candidate genes were highly interconnected during maize mesocotyl and coleoptile elongation at different sowing depth environments (Figure 6C).

### 2.7. Gene Expression Validation by qRT-PCR

To further verify the reliability of the candidate genes, five genes were randomly selected for relative expression level analysis in mesocotyl and coleoptile of both W64A and K12 at 3 cm and 20 cm sowing depths. Our results showed that the expression of these candidate genes varied significantly (*p* < 0.05) in the mesocotyl and coleoptile of both parents at 3 cm and 20 cm sowing depths, which was consistent with the RNA-Seq data (Figure 6D).

## 3. Discussion

### 3.1. Adaptive Changes of Maize in Response to Deep-Sowing Stress

In long-term agricultural practice, deep-sowing is a highly effective strategy for improving the resistance of various crops to drought stress, cold damage, and lodging [6,21,27,28,29,30], and the emergence of seedlings is directly related to cooperative elongation of mesocotyl and coleoptile [3,7,14,31]. In this study, we found that with an increase in sowing depth from 3 cm to 20 cm, the RAT of two parents, F_1_ hybrid, and F_2:3_ population all decreased, their mesocotyls and coleoptiles significantly elongated, and seedlings were slender and weaker (Table 1), which supported previous observations [8,19,21]. Liu et al. [20] also reported that maize germination ability at 12.5 cm sowing depth was correlated with three tissue lengths in the order of mesocotyl > plumule > coleoptile. Pearson pairwise correlation in this study showed that the correlation coefficients between RAT and other traits were MESL + COLL > MESL/COLL > MESL > COLL > SDL at both 15 cm and 20 cm depths (Figure 2C). These findings revealed that seeds sown at the deeper soil layer completed their germination by promoting both mesocotyl and coleoptile elongation. Therefore, the length of mesocotyl and coleoptile can serve as reliable indicators or standards for evaluating maize deep-sowing tolerance. However, despite showing clear elongation and expansion of mesocotyl cells at 20 cm sowing depth compared to 3 cm, some mesocotyl cells appeared to undergo severe programmed cell death due to deep-sowing stress in W64A and K12 [13]. It is therefore important to ensure the synergistic development of mesocotyl and coleoptile when breeding elite maize varieties for deep-sowing stress tolerance.

The elongation of mesocotyl and coleoptile of maize at deep-sowing stimulations can be closely related to some physiological responses. A large amount of evidence has been confirmed that phytohormones are crucial internal regulators that affect the length of mesocotyl and coleoptile. Zhao and Zhong [3] reported that 20 cm deep-sowing stress increased indole-3-acetic acid (IAA), Cis-zeatin (Cis-ZT), trans-zeatin (Trans-ZT), 24-epibrassinolide (EBR), and abscisic acid (ABA) accumulation in both mesocotyl and coleoptile of four maize genotypes. These phytohormones are vital for plant development in which IAA promotes cell enlargement of mesocotyl [32], ABA promotes cell elongation of hypocotyl [33], brassinosteroid (BR) promotes cell elongation and division of both mesocotyl and coleoptile [13,14], and cytokinin (CTK) promotes cell division of mesocotyl [34]. In contrast, Feng et al. [35] found that light irradiation inhibited mesocotyl elongation of rice with a concomitant increase in jasmonic acid (JA) level, and a decrease in IAA and gibberellin 3 (GA_3_) as well as Trans-ZT contents. It is likely that JA was involved in light-dependent regulation of mesocotyl elongation [35]. The mutants impaired in JA biosynthesis had a longer mesocotyl phenotype confirmed the negative role of JA in mesocotyl elongation [36]. Lignin is the main component of secondary cell walls and is directly related to the degree of cell wall elasticity. Cell wall relaxation and cell elongation are negatively affected by increasing lignin levels [37]. Phenylalanine ammonia-lyase (PAL), cinnamyl alcohol dehydrogenase (CAD), trans-cinnamate 4-monooxygenase (C4H), 4-coumarate-CoA ligase (4CL), and peroxidase (POD) all play major roles in lignin biosynthesis [38]. PAL is the first enzyme of the general phenylpropanoid pathway that deaminated phenylalanine to trans-cinnamic acid [39]. During cell wall lignification, the H_2_O_2_ induced POD catalyzed the oxidative dehydrogenation of monolignols (hyproxyphenyl-, guaiacyl-, and syringyl-unit), which led to free radical polymerization to produce lignin [40]. Zhao and Zhong [3] showed that the decrease in PAL, POD, superoxide dismutase (SOD), catalase (CAT), and ascorbate peroxidase (APX) activities in mesocotyl and coleoptile under 20 cm deep-sowing stress reduced lignification, which subsequently accelerated mesocotyl and coleoptile elongation in maize. Targeted metabolomics using ultra-high-performance liquid chromatography-tandem mass spectrometry (UPLC-MS/MS) also showed that some intermediate metabolites of lignin biosynthesis, including *p*-coumaraldehyde, *p*-coumaryl alcohol, and sinapaldehyde, in mesocotyl of two maize inbred lines were down-regulated at 20 cm of the deep-sowing environment [41].

### 3.2. Pleiotropic QTLs Related to Multiple Deep-Sowing Tolerant Traits in Maize

To determine whether the genetic basis of different deep-sowing tolerant traits is similar or distinct, we compared six deep-sowing tolerant traits with their respective QTLs. We identified 33 QTLs for RAT, MESL, COLL, MESL + COLL, MESL/COLL, and SDL in the F_2:3_ population at three sowing depth environments via CIM (Figure 3; Appendix A). For these identified QTLs, only MESL/COLL had non-additive effects (Appendix A), while five other traits had both additive and non-additive effects (Appendix A). The breeding scheme should therefore be designed to utilize both additive and non-additive effects as well as general and specific combining abilities to improve the deep-sowing tolerance of maize. In parallel, we further mapped 13 major QTLs associated with these six traits, and four of them showed pleiotropic effects on two or more deep-sowing tolerant traits, resulting in high effects to PVE (Appendix A). For example, the major QTL in Bin 1.09 of umc2047–bnlg1597 region was associated with MESL, COLL, and MESL + COLL, which individually explained 2.89–14.12% of PVE at a single sowing environment (Appendix A). Its vicinity, i.e., umc1085–umc1968 region, a QTL for RAT and SDL at 10 cm and 20 cm environments, was identified, which explained 7.19–7.27% of PVE [18]; near chr01.2532.5 marker, a QTL for germination rate at 12.5 cm sowing depth, was also detected, which explained 3.32% of PVE [20]; the PZE-101226516–PZE-101229026 region mapped a QTL for seed vigor index, and near PZE-101221874 marker, and a QTL for germination index was found [42]; and in a genome-wide association study (GWAS), a QTL for seed imbibition volume was identified near the Chr.S_262721257 marker [43]. These results all indicated that a pleiotropic QTL in the Bin 1.09 region may be a key locus regulating seed vigor, germination ability, and seedling establishment under deep-sowing stress in maize, which could thus be a target in a breeding scheme.

The major QTL in Bin 4.08 of the umc1612–umc1313 region was associated with MESL + COLL and MESL/COLL, which individually explained 4.08–10.69% of PVE at a single-sowing environment (Appendix A). In the same interval of Bin 4.08, Zhang et al. [18] identified a QTL for SDL in the bnlg1444–bnlg2162 region at both 10 cm and 20 cm depths (PVE was 9.35% and 9.08%, respectively), a QTL for COLL in the umc2041–umc1313 region at 20 cm depth (PVE was 16.01%), and a QTL for MESL in the umc1313–umc2287 region at 20 cm depth (PVE was 4.19%). Thus, we predicted that the candidate genes in the Bin 4.08 region predominantly participated in cooperative development of mesocotyl and coleoptile in maize.

The major QTL in Bin 6.01 of the umc2311–umc2196 region was associated with MESL and MESL + COLL, which individually explained 5.06–10.03% of PVE at a single-sowing environment (Appendix A). This Bin region also contained multiple QTLs related to maize plant height [44,45], ear height [44], under-ear internode length [46], and internode length above the uppermost ear [47] across multiple mapping populations under various environments. Dilday et al. [48] demonstrated that mesocotyl elongation in rice was stably inherited from generation to generation in semi-dwarf varieties, implying a strong genetic basis of this trait. These results suggested that mesocotyl development may be closely related to plant height at maturity in maize, which indicated the feasibility of manipulating both traits simultaneously for longer mesocotyl and desired plant height. Multiple alleles that control both long mesocotyl and dwarfness can be aggregated to develop an elite form of maize in the future.

Additionally, the major QTL in the Bin 7.02 (umc1585–umc2526) region controlled both MESL and COLL, which individually explained 3.67–14.17% of PVE at a single-sowing environment (Appendix A). Previous studies validated pleiotropic QTLs for ear row number, ear weight, ear length, and kernel number per row in the Bin 7.02 region [49,50]. So, we speculated that Bin 7.02 region may be an important QTL cluster, which simultaneously regulated maize deep-sowing tolerant traits and yield-related traits.

### 3.3. Candidate Genes in Major QTLs and MQTLs Regions

The projection of original QTLs on the consensus map enabled us to identify 17 MQTLs for deep-sowing tolerant traits (Table 4; Figure 5); among those, nine major QTLs identified two or more sowing environments that were contained in six MQTLs regions (Table 4). These data showed that the meta-analysis method could map stable QTLs (i.e., major QTLs identified under two or more environments), which off-set the limitation of traditional QTL mapping approaches. Most of the QTLs were environment-dependent loci that might not be effective in different environmental conditions and difficult to utilize in breeding [51]. Furthermore, the 50 potential candidate genes identified in the above major QTLs and MQTLs regions via RNA-Seq in mesocotyl and coleoptile of both parents at two sowing depths (Figure 5 and Figure 6B; Table 4 and Appendix A) might all be useful in developing elite maize lines/varieties. 

It is well known that the circadian clock is an endogenous timekeeping system, which can integrate various cues to regulate plant physiological functions for adaptation to changing environmental conditions and thus ensure optimal plant growth and development [52]. In an earlier transcriptome study, eight differentially expressed genes (DEGs) for *phytochrome A* and *B* (*PHYA* and *PHYB*) were down-regulated in both mesocotyl and coleoptile of two maize genotypes under 20 cm sowing depth stimulation [7]. The end-of-day far-red light (EOD-FR) could reduce ABA level in mesocotyl of both wild type and *phyB1 phyB2* double mutants of maize, which suggested a FR-mediated but PHYB-independent regulation of ABA accumulation [53]. This phenomenon indicated a complex interplay of light and phytohormones in elongation response. In this study, we identified six candidate genes involved in the circadian clock using gene ontology (GO) annotation analysis (Appendix A), i.e., *GRMZM2G124532* (*PHYB1*) in MQTL1–2 region, *GRMZM2G157727* (*PHYA1*) and *GRMZM2G057935* (*PHYC1*) in MQTL1–4 region, *GRMZM2G092174* (*PHYB2*) in MQTL9–1 region that were involved in the detection of visible light (GO:0009584) and red and far-red light phototransduction (GO:0009585); *GRMZM2G107499* for Protein LNK1 in MQTL1–2 region, mainly participated in circadian rhythm (GO:0007623); and *GRMZM2G003501* encoded 3-ketoacyl-CoA synthase in MQTL4–1 region, which regulated the response to cold (GO:0009409) and the response to light stimulus (GO:0009416). 

We identified 27 candidate genes associated with multiple phytohormones biosynthesis and signal transduction in nine MQTLs regions. Related to brassinosteroids, *GRMZM2G065635* encoded brassinosteroid synthesis 1 in the MQTL1–1 region and *GRMZM2G103773* was a *brassinosteroid-deficient dwarf 1* in the MQTL1–1 region, which regulated microtubule bundle formation (GO:0001578), brassinosteroid homeostasis (GO:0010268), brassinosteroid biosynthetic process (GO:0016132), jasmonic acid mediated signaling pathway (GO:0009867), and leaf development (GO:0048366). The knockout of the *GRMZM2G065635* ortholog in *Arabidopsis* led to a dwarfed phenotype and severe growth and development defects [54]. *GRMZM2G455658* was a *Nana2*-like 1 in MQTL4–1 region and was consistent with our previous observation [21], which participated in lignin metabolic process (GO:0009808), unidimensional cell growth (GO:0009826), plant-type secondary cell wall biogenesis (GO:0009834), and brassinosteroid biosynthetic process (GO:0016132). Ten auxin-responsive protein (SAUR) were identified in MQTL1–4, MQTL3–1, MQTL4–1, MQTL5–1, and MQTL7–1 regions, that regulated response to auxin (GO:0009733). A previous study reported that *AtSAUR24* could promote cell expansion and hypocotyl growth in *Arabidopsis* [55]. Five candidate genes responsible for cytokinin signal transduction, i.e., *GRMZM2G040736* (cytokinin response regulator 1; in MQTL2–1), *GRMZM2G404443* (cytokinin dehydrogenase 6 precursor; in qSDL–Ch.1–2), *GRMZM2G022904* (cytokinin hydroxylase; in MQTL7–1), *GRMZM2G126834* (ARR-B-transcription factor 1; in MQTL9–2), and *GRMZM2G122340* (cytokinin oxidase 11; in MQTL10–1) were identified, and GO analysis showed that they were associated with cytokinin dehydrogenase activity (GO:0019139), cytokinin metabolic process (GO:0009690), trans-zeatin biosynthetic process (GO:0033466), and cytokinin-activated signaling pathway (GO:0009736). *GRMZM2G065928* was a putative abscisic acid 8-hydroxylase 4 in MQTL7–1 and was involved in abscisic acid catabolic process (GO:0046345), brassinosteroid homeostasis (GO:0010268), brassinosteroid biosynthetic process (GO:0016132), and multicellular organism development (GO:0007275). *GRMZM2G102163* found in the MQTL4–1 region encoded receptor-like serine/threonine-protein kinase ALE2, which was involved in abscisic acid-activated signaling pathway to regulate cell differentiation (GO:0030154), cuticle development (GO:0042335), and plant organ formation (GO:1905393). In the MQTL4–1 region, there was *GRMZM2G017852* (U-box domain-containing protein 44) played ubiquitin-protein transferase activity (GO:0004842) and ABA biosynthesis (GO:0010115), and *GRMZM2G102216* was a glutathione S-transferase family protein, which displayed glutathione transferase activity (GO:0004364) and auxin-mediated signaling pathways (GO:0010930). *GRMZM2G044358* was a GA3-oxidase 2 isoform X1 in MQTL6–2 region, and showed gibberellin 3-beta-dioxygenase activity (GO:0016707), and *GRMZM2G143328* in MQTL3–2 and *GRMZM2G305856* in MQTL6–1 were MYB transcription factors. Du et al. [56] demonstrated that *ZmMYB59* played a negative regulatory role in maize seed germination in deep soil and the regulation was associated with the GA signaling pathway. The *AP2-EREBP* gene (*GRMZM2G174834*) in the MQTL4–2 region was involved in ethylene signaling [8]. *GRMZM2G403620* was a protein rough sheath 2 (RS2) in the MQTL3–1 region and showed response to auxin (GO:0009733), a response to gibberellin (GO:0009739), a response to salicylic acid (GO:0009751), and a response to jasmonic acid (GO:0009753), leaf morphogenesis (GO:0009965), and leaf formation (GO:0010338). Phelps-Durr et al. [57] showed that the *RS2* in maize and its *Arabidopsis* orthologue *ASYMMETRIC LEAVES1* (*AS1*) could form a protein complex by interacting with a homologue of the chromatin-remodeling protein HIRA, which played a direct role in *knox* gene repression and determinacy during leaf formation. 

Lignin biosynthesis involves the synergistic action of several enzyme systems, including PAL, tyrosine ammonia-lyase (TAL), POD, CAT, and ferulate 5-hydroxylase (F5H) as well as H_2_O_2_ signaling [13]. In the MQTL1–2 region, we identified *GRMZM2G326335* encoding PAL10 and showing phenylalanine ammonia-lyase activity (GO:0045548), L-phenylalanine catabolic process (GO:0006559), and a cinnamic acid biosynthetic process (GO:0009800). Previously, Oh et al. [58] found that *PAL1* and *PAL2* were related to lignin biosynthesis in *Arabidopsis*. The knockdown of *PAL*-isoform by RNA interference in grass *Brachypodium distachyon* decreased 43% of the lignin content in the cell wall of stems, and it delayed plant development and root growth [59]. C4H belongs to the cytochrome P450 monooxygenase family and catalyzed the carboxylation of *p*-coumaric acid to form caffeic acid [60]. We identified a C4H family gene (*GRMZM2G167986*) in MQTL1–4 region that might be involved in lignin biosynthesis. We also identified three *POD* genes (i.e., *GRMZM2G040638*, *GRMZM2G107228*, and *GRMZM2G450233*) in three MQTLs regions. In GWAS, Chen et al. [61] identified *GRMZM2G035506* (*POD7*), which was located near the PZE-105098349 marker and regulated the growth of deep-sowing maize seedlings. Leonard et al. [62] proposed that the spatial control of lignin chemistry depends on different combinations of laccase (LAC) with nonredundant activities immobilized in specific cell types and cell wall layers. In this study, we also identified two *LAC* genes (i.e., *GRMZM2G447271* and *GRMZM2G336337*) in MQTL8–1 region that were annotated to the lignin catabolic process (GO:0046274).

Cell wall organization is also an important factor affecting cell elongation and expansion. *GRMZM2G419267* was identified in the MQTL1–3 region encoding beta-2-xylosyltransferase and having xylosyltransferase activity (GO:0042285) to regulate primary cell wall biosynthesis. *GRMZM2G059212* related to callose synthase 3 was identified in the MQTL5–1 region and showed regulation of cell shape (GO:0008360) and cell wall organization (GO:0071555). *GRMZM2G112984* annotated to pectinesterase activity (GO:0030599) identified in qRAT–Ch.7–1 and controlled cell wall modification (GO:0042545). Kang et al. [63] showed that *cell wall invertase (INCW2)* in maize was required for normal development of labyrinth-like wall-in-growth to increase the plasma membrane area in developing seeds, and *GRMZM2G174249*, an ortholog of *Arabidopsis INCW4*, was identified in the MQTL3–1 region.

It is generally believed that the amounts of osmotic solutes, including sugar in cells, determines the force of water uptake, thus controlling the rate of cell growth. We identified *GRMZM2G130043* (Starch synthase V) and *GRMZM2G045171* (sucrose synthase 6) in the MQTL4–1 region, which has starch synthase activity (GO:0009011) and sucrose synthase activity (GO:0016157), respectively. It is consistent with previous results that demonstrated the suppression of sugar accumulation in maize mesocotyl and coleoptile cells as well as their growth when etiolated seedlings were subjected to white light irradiation [64]. *GRMZM2G050705* was a putative glycosyltransferase family protein 28 in qMESL–Ch.6–1 involved in transferase activity (GO:0016740), and it controlled the dolichol-linked oligosaccharide biosynthetic process (GO:0006488).

Moreover, of the two candidate genes involved in the antioxidant enzyme system, *GRMZM2G088212* encoded catalase 1 (CAT1) and showed catalase activity (GO:0004096), peroxidase activity (GO:0004601), circadian rhythm (GO:0007623), response to abscisic acid (GO:0009737), and response to salicylic acid (GO:0009751) identified in the MQTL5–1 region; and *GRMZM2G460406* is a putative ascorbate peroxidase 9 found in qMESL/COLL–Ch.4–1 region and involved in the response to reactive oxygen species (GO:0000302), L-ascorbate peroxidase activity (GO:0016688), and the hydrogen peroxide catabolic process (GO:0042744).

## 4. Materials and Methods

### 4.1. Plant Materials

The 346 F_2_ individuals derived from the W64A × K12 F_1_ hybrid were used to construct the genetic linkage map [21]; the 346 F_2:3_ population families were obtained by self-crossing of all F_2_ plants. These plants were used to measure six deep-sowing tolerant traits at the culture of 3 cm, 15 cm, and 20 cm sowing depths for 10 days and the data were used to perform QTL analysis. The F_2_ and F_2:3_ populations were developed at the Longxi experimental stations, Gansu, China (34.97° N, 104.40° E, 2.074 m altitude). The female parent, W64A, originated from the Lancaster heterotic group with strong deep-sowing tolerance, while the male parent, K12, originated from the Tang Si Ping Tou heterotic group with intolerance [21,65]. 

### 4.2. Measurement of Six Deep-Sowing Tolerant Traits

Seeds of W64A, K12, F_1_ hybrid, and 346 F_2:3_ families were sown at 3 cm, 15 cm, and 20 cm sowing depths in PVC tubes (height: 50 cm; diametaer: 17 cm) containing vermiculites. The seeds were first soaked in distilled water for 24 h at 20 °C, and 30 soaked seeds were then sown in the PVC tubes with 50 cm of evenly-mixed vermiculite. Plants were grown in a greenhouse (22 ± 1 °C constant temperature, light for 12 h per day, 65% relative humidity) for 10 days. 20 mL distilled water was added into the vermiculite at 2-day intervals. Each sowing depth treatment was replicated three times. Ten days after germination, RAT, MESL, COLL, MESL + COLL, MESL/COLL, and SDL of ten seedlings were measured at each sowing environment.

For six deep-sowing tolerant traits of both parents, F_1_ hybrid, and 346 F_2:3_ families at three sowing environments, factorial analysis of variance (ANOVA), principal component analysis (PCA), and multiple linear regression analysis were performed using IBM-SPSS Statistics v.19.0 (SPSS Inc., Chicago, IL, USA; http://www.ibm.com/products/spss-statistics; accessed on 10 July 2022). The broad-sense heritability ($H_{B}^{2}$) and genotype × environment interaction heritability ($H_{GE}^{2}$) values of the above six traits in the F_2:3_ population were calculated as follows [66,67]:*H_B_*^2^ = σ_g_^2^/(σ_g_^2^ + σ_ge_^2^/n + σ_ε_^2^/nr),(1)
*H_GE_*^2^ = (σ_g_^2^/n)/(σ_g_^2^ + σ_ge_^2^/n + σ_ε_^2^/nr)(2)
where σ_g_^2^ was genotypic variance, σ_e_^2^ was environmental variance, σ_ε_^2^ was error variance, σ_ge_^2^ was variance of genotype × environment interaction, n (n = 3) was the number of sowing environments, and r (r = 10) was the number of replications. The coefficient of variation (CV) of these traits at each environment was estimated as follows [21]:(3)$$\mathrm{CV}=\delta /\bar{x}\times 100\%$$
where $\bar{x}$ was average value of corresponding traits in the F_2:3_ population at each environment, and δ was standard deviation. The heterosis of all traits were evaluated at all environments as follows [21]:
HI = F_1_/MP × 100%(4)
RH = (F_1_ − MP)/F_1_ × 100%(5)
MH = (F_1_ − MP)/MP × 100%(6)
OH = (F_1_ − P_H_)/P_H_ × 100%(7)
ARR = (F_2:3_ − F_1_)/F_1_ × 100%(8)
where MP was the average value of corresponding trait of both parental lines, and P_H_ was highest value of corresponding trait in the parental line. The Pearson pairwise correlation among all traits at different sowing environments were prepared using the Genescloud tool (https://www.genescloud.cn; accessed on 12 December 2022). 

### 4.3. Genetic Linkage Map Construction and QTL Analysis

The polymorphic SSRs were screened out between W64A and K12 from the MaizeGDB website (http://www.maizegdb.org/; accessed on 22 January 2020) that were used to distinguish genotypes of 346 F_2_ individuals and draw the genetic linkage map through JoinMap v.4.0 (https://www.kyazma.nl/index.php/JoinMap/; accessed on 16 June 2021) [21]. The identification of QTLs for six deep-sowing tolerant traits at three sowing depths environments were performed using Windows QTL Cartographer software v.2.5 (https://brcwebportal.cos.ncsu.edu/qtlcart/WQTLCart.htm; accessed on 21 July 2022) with the composite interval mapping (CIM) approach. For CIM, model 6 of the Zmapqtl module was used to identify QTLs. Window size was 10 cm, and cofactors were selected through forward and backward regressions with the in-and-out thresholds at a *p* < 0.05. We estimated a genome-wide critical threshold value for an experimental type I error rate of 0.05 using 1000 random permutations [68]. The QTL was named according to the modifying nomenclature of McCouch et al. [69]. We used |dominance (d)/additive (a)| value to estimate the gene action of every QTL: additive (A; |d/a| = 0.00~0.20), partial-dominance (PD; |d/a| = 0.21~0.80), dominance (D; |d/a| = 0.81~1.20), and over-dominance (OD; |d/a| > 1.20) [70].

### 4.4. Consensus Linkage Map Construction and Meta-Analysis

The original QTL mapping information from three studies for seven deep-sowing tolerant traits at five sowing depth environments were selected from the public databases: NCBI (https://www.ncbi.nlm.nih.gov/; accessed on 23 November 2022), MaizeGDB (http://www.maizegdb.org/; accessed on 23 November 2022), and CNKI (https://www.cnki.net/; accessed on 23 November 2022). These QTLs information mainly included original chromosomal position, LOD scores, confidence interval (CI), and proportion of PVE. Subsequently, the consensus linkage map was developed via BioMercator v.4.2.3 software (https://urgi.versailles.inra.fr/Tools/BioMercator-V4; accessed on 3 December 2022) and this map was compared with the IBM2 2008 Neighbors Map Frame 6 reference map (https://www.maizegdb.org/data_center/map; accessed on 1 December 2022). If the CI for QTL position was not available in the published literature, we then estimated a 95% CI as follows [71]:
CI = 163/(N × R^2^)(9)
CI = 530/(N × R^2^)(10)
where N was the size of the mapping population and R^2^ was the phenotypic variance. Equation (9) was appropriate for recombinant inbred lines (RILs) population, and Equation (10) was appropriate for the F_2_ or backcross population. After we identified the QTLs in the original populations with a homothetic function and successfully projected the QTLs on the consensus maps, we used algorithms for meta-analysis to identify the number and position of the MQTLs by BioMercator v.4.2.3. According to the physical diatance of markers, we projected the MQTLs and major QTLs on the physical map B73 RefGen_v3 (https://www.maizegdb.org/genome/assembly/B73%20RefGen_v3; accessed on 10 December 2022).

### 4.5. Candidate Genes Identification via RNA-Seq

To accurately identify candidate genes within corresponding major and MQTLs regions, the RNA-Seq was performed in mesocotyl [13] and coleoptile [14] of both W64A and K12 at 3 cm and 20 cm sowing environments using an Illumina Nova Seq PE150 sequencer at Nanjing Genepioneer Biotechnologies Company (Nanjing, Jiangsu, China). The transcriptome quality was shown in Appendix A. The expression levels of merged transcripts were counted and FPKM values were calculated as by Trapnell et al. [72]. The FPKM expression differences of genes in mesocotyl and coleoptile of both parents at two sowing environments and their GO annotation (https://agbase.arizona.edu/; accessed on 16 December 2022) were combined to identify candidate genes in above major QTLs and MQTLs regions. The physical map of all major QTLs, MQTLs, and candidate genes was constructed using MapInspect 1.0 (https://mapinspect.software.informer.com/; accessed on 20 December 2022), and the physical length of each chromosome was obtained through B73 RefGen_v3 genome (https://www.maizegdb.org/genome/assembly/B73%20RefGen_v3; accessed on 20 December 2022). The expression profiles of candidate genes were performed using TBtools software (https://github.com/CJ-Chen/TBtools/releases; accessed on 21 December 2022) [73,74].

### 4.6. qRT-PCR Analysis

0.5 μg RNA was reverse-transcribed to synthesise first-stand cDNA using HiScript^®^Q RT SuperMix for qPCR (Vazyme, China), according to the manufacturer’s protocol. qRT-PCR was conducted on quantum Studio 5 real-time PCR system (Thermo Fisher Scientific, Waltham, MA, USA) using super real premix plus (SYBR Green) (Tiangen, Shanghai, China). The five candidate genes were randomly selected and their specific primers were designed with Primer3web v4.1.0 (https://primer3.ut.ee/ accessed on 22 December 2022) (Appendix A). The relative expression level of the genes were calculated by the 2^−ΔΔCt^ method, with *GRMZM2G126010* (*Actin–1*) as an internal reference gene [65].

## 5. Conclusions

In summary, deep-sowing has been proposed as an effective strategy to deal with drought stress and cold damage during maize seed germination, and the cooperative elongation ability of mesocotyl and coleoptile determined maize deep-sowing tolerance. Here, we systematically analyzed the heterosis and 33 QTLs for six deep-sowing tolerant traits in a population of 346 F_2:3_ families at three sowing depths environments. Among these QTLs, 12 of 13 major QTLs were stably identified under two or more sowing environments, except for qMESL–Ch.1–2, and four major QTLs in Bin 1.09, Bin 4.08, Bin 6.01, and Bin 7.02 supported pleiotropy for multiple deep-sowing tolerant traits. For the first time, the meta-analysis method was used to investigate the genetic background of deep-sowing tolerance in maize, which identified a total of 17 MQTLs controlling seven deep-sowing tolerant traits. Nine of the major QTLs identified in the F_2:3_ population were overlapped with six MQTLs. RNA-Seq analysis in mesocotyl and coleoptile of both parents at 3 cm and 20 cm sowing environments identified 50 candidate genes in all major QTLs and MQTLs regions that were mainly involved in the circadian clock, phytohormones biosynthesis and signal transduction, lignin biosynthesis, cell wall organization formation and stabilization, sucrose and starch metabolism, and the antioxidant enzyme system. These genes form highly interconnected networks, which may elucidate the complex molecular mechanism of maize deep-sowing tolerance (Figure 7). Findings of this study may facilitate a future marker-assisted QTL pyramiding/introgression breeding scheme to improve maize deep-sowing tolerance.

## Figures and Tables

**Figure 1 ijms-24-06770-f001:**
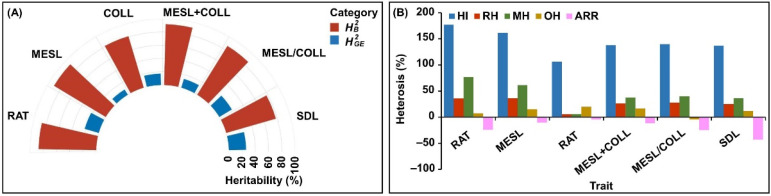
The broad-sense heritability ($H_{B}^{2}$) and genotype × environment interaction heritability ($H_{GE}^{2}$ ), and heterosis for six deep-sowing traits including emergence rate (RAT), mesocotyl length (MESL), coleoptile length (COLL), total length of mesocotyl and coleoptile (MESL + COLL), length ratio of mesocotyl to coleoptile (MESL/COLL), and seedling length (SDL). Multi-angle radar map of $H_{B}^{2}$ and $H_{GE}^{2}$ for the six traits was prepared using the Genescloud tool (https://www.genescloud.cn; accessed on 12 December 2022) (**A**). Heterosis, including F_1_ heterosis index (HI), relative heterosis (RH), mid-parent heterosis (MH), over-parent heterosis (OH), and F_2:3_ advantage reduction rate (ARR) of these six traits (**B**).

**Figure 2 ijms-24-06770-f002:**
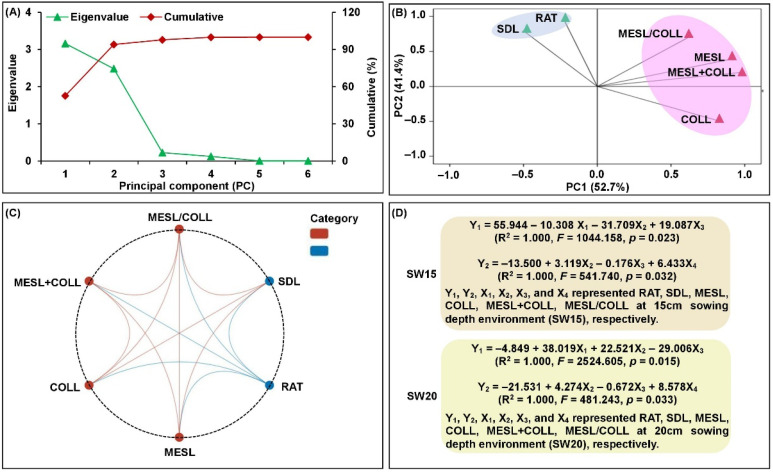
Framework relations based on principal component analysis (PCA), Pearson pairwise correlations, and multiple linear regression analysis among six deep-sowing tolerant traits, emergence rate (RAT), mesocotyl length (MESL), coleoptile length (COLL), total length of mesocotyl and coleoptile (MESL + COLL), length ratio of mesocotyl to coleoptile (MESL/COLL), and seedling length (SDL), at three sowing depth (3 cm, 15 cm, and 20 cm) environments. Eigenvalues and cumulative variance of principal components (PCs) (**A**). The effects of six deep-sowing tolerant traits in PC1 and PC2 (**B**). Interactive ring correlation diagram among six deep-sowing tolerant traits at the sowing depths of both 15 cm and 20 cm, prepared using the Genescloud tool (https://www.genescloud.cn; accessed on 12 December 2022) (**C**). Multiple optimal linear regression equations of six deep-sowing tolerant traits at 15 cm and 20 cm treatments were constructed using IBM-SPSS Statistics v.19.0 (SPSS Inc., Chicago, IL, USA; http://www.ibm.com/products/spss-statistics; accessed on 10 July 2022) (**D**).

**Figure 3 ijms-24-06770-f003:**
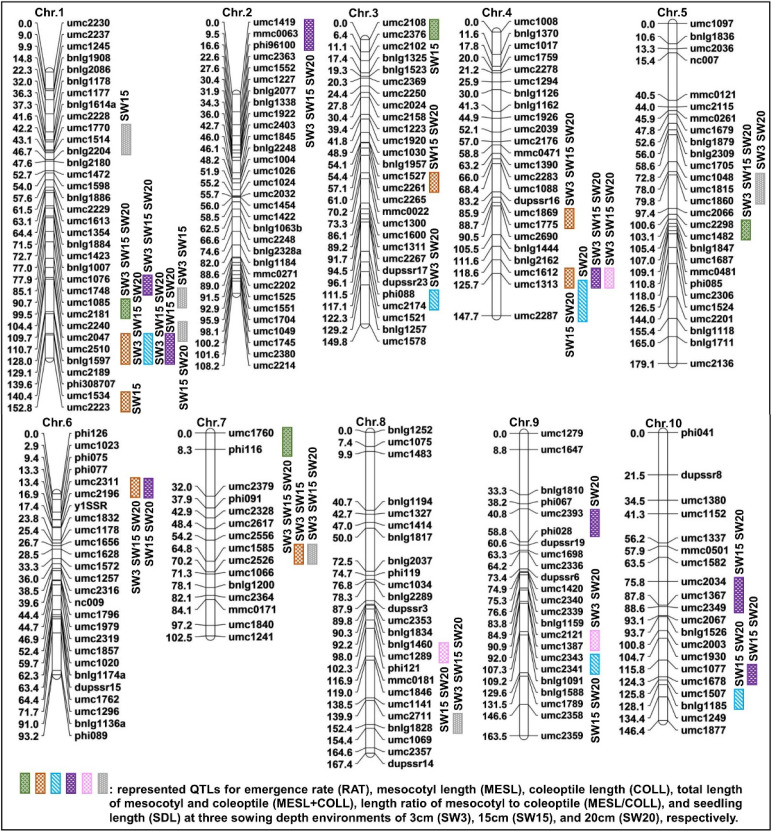
Genetic linkage map construction and QTL mapping for six deep-sowing tolerant traits, emergence rate (RAT), mesocotyl length (MESL), coleoptile length (COLL), total length of mesocotyl and coleoptile (MESL + COLL), length ratio of mesocotyl to coleoptile (MESL/COLL), and seedling length (SDL) in the F_2:3_ population at three sowing depths of 3 cm (SW3), 15 cm (SW15), and 20 cm (SW20), respectively.

**Figure 4 ijms-24-06770-f004:**
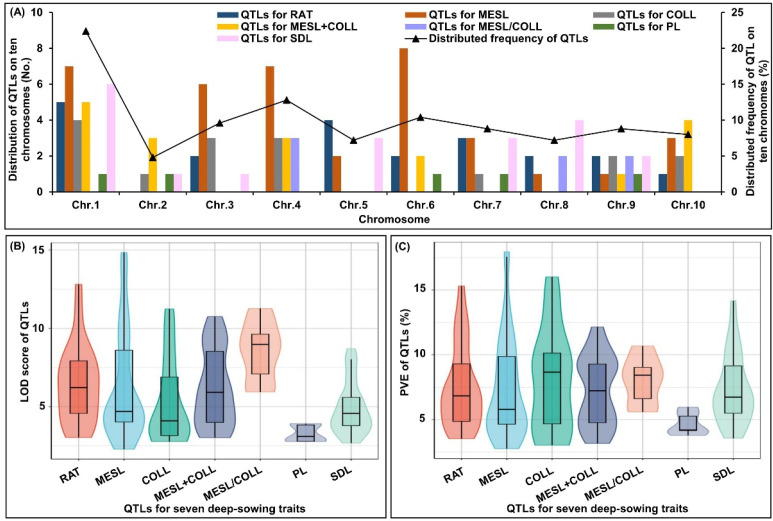
Summary of original QTLs information for seven deep-sowing tolerant traits, emergence rate (RAT), mesocotyl length (MESL), coleoptile length (COLL), total length of mesocotyl and coleoptile (MESL + COLL), length ratio of mesocotyl to coleoptile (MESL/COLL), plumule length (PL), and seedling length (SDL). Distribution of QTLs on ten chromosomes (**A**), logarithm of odds (LOD) scores of QTLs (**B**), and phenotypic variance (PVE) of QTLs (**C**) for seven deep-sowing tolerant traits.

**Figure 5 ijms-24-06770-f005:**
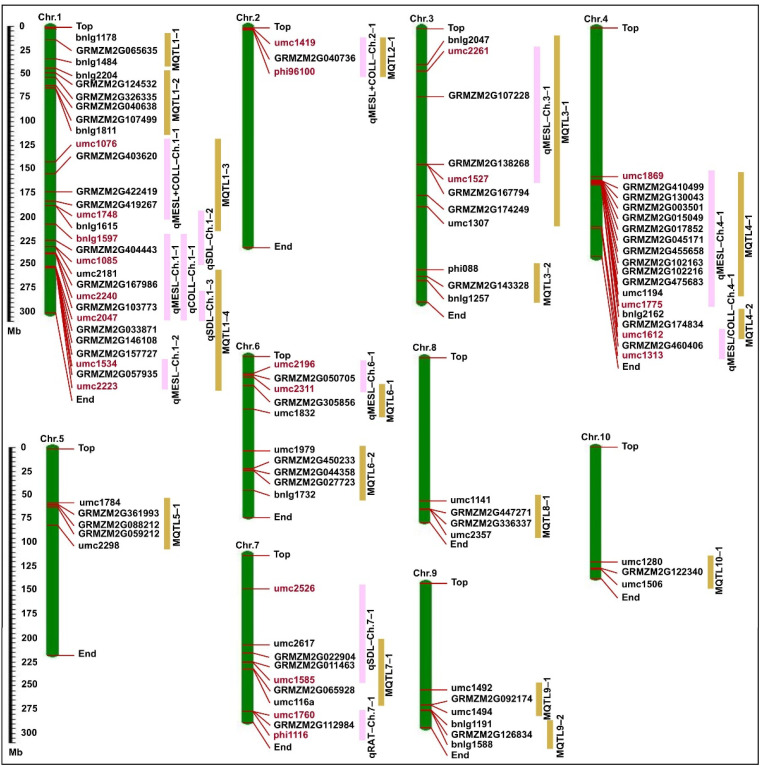
The physical map was constructed according to B73 RefGen_v3 physical map (https://www.maizegdb.org/genome/assembly/B73%20RefGen_v3; accessed on 10 December 2022); the distributions of major QTLs, meta-QTLs (MQTLs), and candidate genes in the physical map were performed using MapInspect 1.0 software (https://mapinspect.software.informer.com; accessed on 13 December 2022). Pink pillars represent corresponding major QTLs regions in our F_2:3_ population (vermilion markers represent the two side markers of corresponding major QTLs), and earthy yellow pillars represent MQTLs regions.

**Figure 6 ijms-24-06770-f006:**
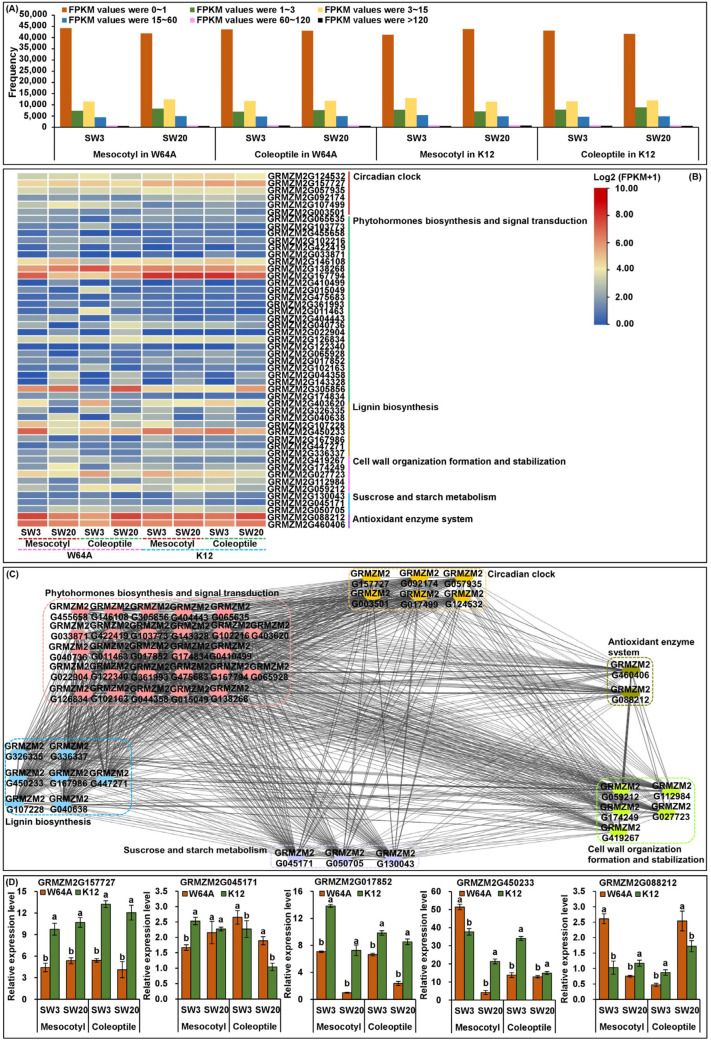
Gene expressions as the fragments per kilobase of transcript per million mapped read (FPKM) values in mesocotyl and coleoptile of both parents (W64A and K12) at 3 cm (SW3) and 20 cm (SW20) sowing depths via RNA-sequencing (RNA-Seq) (**A**). The expression profiles of candidate genes in mesocotyl and coleoptile of both parents that were performed using TBtools software (https://github.com/CJ-Chen/TBtools/releases; accessed on 21 December 2022) (**B**). The interactome networks among all candidate genes were constructed using Cytoscape 3.8.2 (https://cytoscape.org/download.html; accessed on 26 December 2022) (**C**). Relative expression level of five candidate genes in mesocotyl and coleoptile of both parents at SW3 and SW20 sowing environments, and different lowercase letters over mesocotyl or coleoptile of both parents at a single environment indicate significant differences at *p* < 0.05 (ANOVA) (**D**).

**Figure 7 ijms-24-06770-f007:**
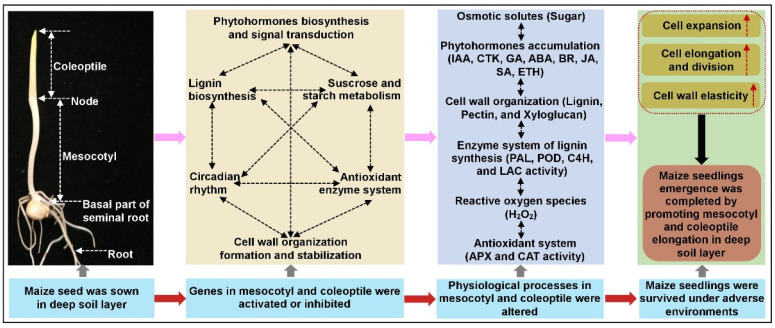
The Molecular networks underlying the deep-sowing stress response of mesocotyl and coleoptile elongation in maize. ABA, abscisic acid; APX, ascorbate peroxidase; BR, brassinosteroid; CAT, catalase; C4H, trans-cinnamate 4-monooxygenase; CTK, cytokinin; ETH, ethylene; GA, gibberellin; IAA, indole-3-acetic acid (IAA); JA, jasmonic acid; LAC, laccase; PAL, phenylalanine ammonia-lyase; POD, peroxidase; SA, salicylic acid. The red dotted arrow indicated that the corresponding processes acted positively.

**Table 1 ijms-24-06770-t001:** Six deep-sowing tolerant traits of two parents, F_1_ hybrid, and F_2:3_ population at three sowing depths.

Trait	Sowing Depth	Female Parent (W64A)	Male Parent (K12)	F_1_ Hybrid (W64A × K12)	F_2:3_ Population (346 Families)				
					Mean ± SD	Range	CV (%)	Skewness	Kurtosis
RAT (%)	SW3	98.67 ± 1.82	96.66 ± 3.34	100.00 ± 0.00	89.18 ± 8.74	53.33–100.00	9.80	0.768	0.361
	SW15	95.32 ± 3.81	6.67 ± 2.36	99.33 ± 1.49	78.21 ± 28.82	20.00–100.00	25.02	−0.097	0.886
	SW20	78.66 ± 1.82	0.00 ± 0.00	92.00 ± 5.05	54.97 ± 19.01	13.33–96.67	34.58	0.290	−0.618
MESL (cm)	SW3	3.82 ± 0.46	2.32 ± 0.29	3.85 ± 0.13	3.74 ± 0.61	1.69–4.79	16.31	0.112	0.110
	SW15	10.28 ± 1.11	3.28 ± 0.11	12.85 ± 0.27	10.95 ± 2.57	2.87–12.62	23.47	0.450	0.301
	SW20	12.70 ± 0.42	5.20 ± 0.13	15.16 ± 0.40	13.09 ± 5.88	3.91–15.05	44.92	0.982	0.337
COLL (cm)	SW3	3.14 ± 0.35	2.94 ± 0.05	3.16 ± 0.12	2.93 ± 0.43	1.18–3.08	14.68	0.327	0.319
	SW15	4.11 ± 0.63	6.20 ± 0.17	5.48 ± 0.38	5.02 ± 1.12	2.25–6.84	22.31	−0.044	0.571
	SW20	4.87 ± 0.31	6.48 ± 0.26	6.12 ± 0.16	6.24 ± 1.75	2.69–6.95	28.04	1.087	0.934
MESL + COLL (cm)	SW3	6.96 ± 0.75	5.26 ± 0.32	7.00 ± 0.12	6.13 ± 1.39	2.94–7.98	22.82	0.578	−0.113
	SW15	14.39 ± 1.45	9.49 ± 0.20	18.32 ± 0.48	15.42 ± 5.52	5.11–19.06	35.80	0.832	0.974
	SW20	17.57 ± 0.24	11.68 ± 0.33	21.28 ± 0.53	19.94 ± 6.71	6.65–23.06	33.65	0.746	0.900
MESL/COLL	SW3	1.22 ± 0.11	0.79 ± 0.09	1.22 ± 0.08	1.24 ± 0.15	1.05–1.73	12.10	0.612	0.015
	SW15	2.54 ± 0.44	0.53 ± 0.02	2.35 ± 0.16	1.38 ± 0.32	1.07–1.87	23.19	0.931	0.798
	SW20	2.62 ± 0.26	0.80 ± 0.03	2.48 ± 0.05	1.64 ± 0.49	1.31–2.20	29.88	0.913	1.017
SDL (cm)	SW3	15.64 ± 0.90	14.12 ± 0.35	16.18 ± 0.83	10.86 ± 1.23	5.73–16.26	11.33	0.511	0.434
	SW15	13.25 ± 0.31	7.58 ± 0.44	15.48 ± 0.48	8.33 ± 3.92	5.27–15.02	47.06	0.385	0.962
	SW20	10.06 ± 0.58	5.18 ± 0.59	11.58 ± 1.07	5.81 ± 1.96	3.93–12.94	33.73	−0.683	−0.790

RAT, emergence rate; MESL, mesocotyl length; COLL, coleoptile length; MESL + COLL, total length of mesocotyl and coleoptile; MESL/COLL, length ratio of mesocotyl to coleoptile; SDL, seedling length; SD, standard deviation; CV, coefficient of variation; SW3, SW15, and SW20 indicate the sowing depth environment of 3 cm, 15 cm, and 20 cm, respectively.

**Table 2 ijms-24-06770-t002:** Pearson correlation coefficient of six deep-sowing tolerant traits between progeny populations (F_1_ hybrid and F_2:3_ population) and parents (W64A and K12).

Item	Progeny Populations					
	RAT	MESL	COLL	MESL + COLL	MESL/COLL	SDL
	F_1_ hybrid (W64A × K12)					
Female parent (W64A)	0.775 *	0.994 ***	0.985 ***	0.994 ***	0.995 ***	0.167
Male parent (K12)	0.806 **	0.903 **	0.928 ***	0.971 ***	−0.311	0.713 *
	F_2:3_ population					
Female parent (W64A)	0.937 ***	0.986 ***	0.976 ***	0.987 ***	0.975 ***	0.220
Male parent (K12)	0.782 *	0.919 ****	0.922 ***	0.965 ***	−0.205	0.890 **

RAT, emergence rate; MESL, mesocotyl length; COLL, coleoptile length; MESL + COLL, total length of mesocotyl and coleoptile; MESL/COLL, length ratio of mesocotyl to coleoptile; SDL, seedling length; */**/*** indicates significant correlation at the *p* < 0.05/*p* < 0.01/*p* < 0.001 probability level, respectively.

**Table 3 ijms-24-06770-t003:** Collection of corresponding mapping datasets of QTLs for seven deep-sowing tolerant traits in maize.

Population		Sowing Depth	Markers Number	Length (cm)	Identified QTLs by Each Sowing Depth Environment							Reference
Type	Size				RAT	MESL	COLL	MESL + COLL	MESL/COLL	PL	SDL	
3681–4 × X178 F_2:3_	221	SW10 SW20	178	1865.5	6	12	3	–	–	–	4	[18]
B73 × Mo17 IBM Syn4 RILs	243	SW12.5	1339	6242.7	7	10	2	–	–	5	7	[19]
W64A × K12 F_2:3_	346	SW3 SW15 SW20	253	1410.6	10	16	10	18	7	–	13	This study

RILs, recombinant inbred lines; RAT, emergence rate; MESL, mesocotyl length; COLL, coleoptile length; MESL + COLL, total length of mesocotyl and coleoptile; MESL/COLL, length ratio of mesocotyl to coleoptile; PL, plumule length; SDL, seedling length; SW3, SW10, SW12.5, SW15, and SW20 indicate the sowing depth environment of 3 cm, 10 cm, 12.5 cm, 15 cm, and 20 cm, respectively.

**Table 4 ijms-24-06770-t004:** The major QTLs identified in the F_2:3_ population derived from W64A × K12, and the candidate genes predicted in corresponding major QTLs and meta–QTLs (MQTLs) intervals.

Trait	Meta–QTL (MQTL)	Major QTLs in This Study	QTLs Number	Bin	Marker Interval	Physical Interval (Mb)	Contig	Candidate Gene	Orthologs
MESL	MQTL1–1	–	2	1.02–1.03	bnlg1178–bnlg1484	14.07–34.92	ctg6–ctg11	GRMZM2G065635	LOC_Os03g12660
COLL, PL, SDL	MQTL1–2	–	4	1.03–1.04	bnlg2204–bnlg1811	45.94–70.81	ctg11–ctg17	GRMZM2G124532	LOC_Os03g19590
								GRMZM2G326335	–
								GRMZM2G040638	LOC_Os03g25330
								GRMZM2G107499	LOC_Os03g27019
MESL + COLL, MESL, SDL	MQTL1–3	qMESL + COLL–Ch.1–1	6	1.05–1.06	umc1076–bnlg1615	143.53–192.99	ctg30–ctg40	GRMZM2G403620	LOC_Os12g38400
								GRMZM2G422419	LOC_Os12g43110
								GRMZM2G419267	LOC_Os08g39380.1
SDL	–	qSDL–Ch.1–2	2	1.06–1.08	umc1748–umc1085	191.89–236.21	ctg40–ctg48	GRMZM2G404443	LOC_Os10g34230
RAT, MESL, COLL, MESL + COLL	MQTL1–4	qMESL–Ch.1–1, qCOLL–Ch.1–1, qSDL–Ch.1–3, qMESL–Ch.1–2	13	1.08–1.10	umc2181–umc2223	237.72–278.97	ctg48–ctg58	GRMZM2G167986	LOC_Os10g26340
								GRMZM2G103773	LOC_Os03g40540
								GRMZM2G033871	LOC_Os03g45850
								GRMZM2G146108	LOC_Os03g45850
								GRMZM2G157727	LOC_Os03g51030
								GRMZM2G057935	LOC_Os03g54084
COLL, MESL + COLL	MQTL2–1	qMESL + COLL–Ch.2–1	4	2.00–2.01	umc1419–phi96100	1.04–2.83	ctg68–ctg68	GRMZM2G040736	LOC_Os04g57720
RAT, MESL, SDL	MQTL3–1	qMESL–Ch.3–1	7	3.04–3.05	bnlg2047–umc1307	30.95–152.88	ctg117–ctg128	GRMZM2G107228	LOC_Os01g22336
								GRMZM2G138268	LOC_Os12g40890
								GRMZM2G167794	LOC_Os12g40900
								GRMZM2G174249	LOC_Os01g73580
COLL, MESL, SDL	MQTL3–2	–	4	3.08–3.09	phi088–bnlg1257	208.61–217.91	ctg134–ctg149	GRMZM2G143328	LOC_Os01g45090
MESL	MQTL4–1	qMESL–Ch.4–1	4	4.06–4.07	umc1869–umc1194	161.53–178.27	ctg182–ctg182	GRMZM2G410499	–
								GRMZM2G130043	LOC_Os02g56320
								GRMZM2G003501	LOC_Os02g49920
								GRMZM2G015049	–
								GRMZM2G017852	–
								GRMZM2G045171	LOC_Os02g58480
								GRMZM2G455658	–
								GRMZM2G102163	LOC_Os02g57080
								GRMZM2G102216	LOC_Os02g56970
								GRMZM2G475683	LOC_Os02g52990
COLL, ESL, MESL + COLL MESL/COLL, SDL	MQTL4–2	–	13	4.08–4.08	bnlg2162–umc1612	185.73–187.53	ctg184–ctg185	GRMZM2G174834	LOC_Os11g03540
MESL/COLL	–	qMESL/COLL–Ch.4–1	3	4.08–4.08	umc1612–umc1313	187.53–224.58	ctg185–ctg189	GRMZM2G460406	LOC_Os08g43560
MESL, SDL, RAT	MQTL5–1	–	6	5.03–5.04	umc1784–umc2298	59.17–84.83	ctg220–ctg225	GRMZM2G361993	LOC_Os06g50040
								GRMZM2G088212	LOC_Os06g51150
								GRMZM2G059212	LOC_Os06g51270
MESL	–	qMESL–Ch.6–1	3	6.01–6.01	umc2311–umc2196	18.71–24.61	ctg262–ctg264	GRMZM2G050705	–
MESL, MESL + COLL	MQTL6–1	–	5	6.01–6.01	umc2311–umc1832	24.61–60.33	ctg262–ctg262	GRMZM2G305856	LOC_Os05g02420
MESL, RAT	MQTL6–2	–	5	6.04–6.05	umc1979–bnlg1732	106.23–151.97	ctg276–ctg287	GRMZM2G450233	LOC_Os05g06970
								GRMZM2G044358	LOC_Os05g08540
								GRMZM2G027723	LOC_Os05g08370
MESL, SDL	MQTL7–1	qSDL–Ch.7–1	5	7.02–7.02	umc2617–umc116a	100.30–127.13	ctg310–ctg317	GRMZM2G022904	LOC_Os09g23820
								GRMZM2G011463	LOC_Os09g26590
								GRMZM2G065928	LOC_Os09g28390
RAT	–	qRAT–Ch.7–1	3	7.06–7.06	umc1760–phi116	174.40–174.61	ctg325–ctg325	GRMZM2G112984	LOC_Os07g49100
SDL	MQTL8–1	–	3	8.06–8.07	umc1141–umc2357	158.93–169.62	ctg361–ctg364	GRMZM2G447271	LOC_Os01g62480
								GRMZM2G336337	LOC_Os01g61160
MESL, COLL, SDL	MQTL9–1	–	5	9.04–9.05	umc1492–umc1494	119.96–136.63	ctg383–ctg387	GRMZM2G092174	LOC_Os03g19590
RAT, PL	MQTL9–2	–	2	9.06–9.07	bnlg1191–bnlg1588	143.19–147.21	ctg389–ctg390	GRMZM2G126834	LOC_Os03g12350
COLL, MESL + COLL	MQTL10–1	–	4	10.04–10.04	umc1280–umc1506	128.12–133.25	ctg413–ctg414	GRMZM2G122340	LOC_Os04g44230

RAT, emergence rate; MESL, mesocotyl length; COLL, coleoptile length; MESL + COLL, total length of mesocotyl and coleoptile; MESL/COLL, length ratio of mesocotyl to coleoptile; PL, plumule length; SDL, seedling length.

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
