# Peer review of "Integrated QTL Mapping, Meta-Analysis, and RNA-Sequencing Reveal Candidate Genes for Maize Deep-Sowing Tolerance"

_ijms, 2023, doi:10.3390/ijms24076770_

Round 1

Reviewer 1 Report

Dear Author

Please find my comments on your manuscript in the pdf attached file.

Author Response

Thank you for your letter of – and for the referee’s comments concerning our manuscript, “Integrated QTL Mapping, Meta-Analysis, and RNA-Sequencing Reveal Candidate Genes for Maize Deep-Sowing Tolerance (ijms-2300221)”. We have carefully studied these comments and have made corresponding corrections to the manuscript, which we describe in detail below. We would like to re-submit the manuscript and that for possible publication on the Special Issue: “Molecular Research in Maizeof International Journal of Molecular Sciences. Thank you very much for your time and consideration.

Editor:

Your manuscript has been reviewed by experts in the field. Please find your manuscript with the referee reports at this link: https://susy.mdpi.com/user/manuscripts/resubmit/c54bdf2e368b9bacf63d8 bdf5b95437b

Thanks for the positive comments of you and all reviewers for our manuscript. As suggested, we have carefully revised and improved the manuscript using the “Track Changes” function of the manuscript at the above link. We then have re-submitted the manuscript within the allotted time.

Thank you for your consideration.

(I) Please revise your manuscript according to the referees’ comments and upload the revised file within 5 days.

Thanks for your positive comments for our manuscript. As suggested, we have carefully revised the manuscript. We then have re-submitted the manuscript within the allotted time.

Thank you for your consideration.

(II) Please use the version of your manuscript found at the above link for your revisions.

 Thanks for the positive comments for our manuscript. As suggested, we have carefully revised and improved the manuscript using the versions of our manuscript at the above link. We then have re-submitted the manuscript.

Thank you for your consideration.

(III) Please check that all references are relevant to the contents of the manuscript.

Thanks for your positive comments for our manuscript. As suggested, we have carefully checked all references to make sure they are relevant to the contents of the manuscript (Lines 771-927). We then have re-submitted the manuscript.

Thank you for your consideration.

(IV) Any revisions made to the manuscript should be marked up using the “Track Changes” function if you are using MS Word/LaTeX, such that changes can be easily viewed by the editors and reviewers.

Thanks for your positive comments for our manuscript. As suggested, we have carefully revised and improved the manuscript using the “Track Changes” function of our manuscript at the above link. We then have re-submitted the manuscript within the allotted time.

Thank you for your consideration.

(V) Please provide a short cover letter detailing your changes for the editors’ and referees’ approval.

Thanks for your positive comments for our manuscript. As suggested, we have carefully revised and improved the manuscript. In addition, we have prepared a detailed response letter to all reviewers for each point, and then have re-submitted the manuscript.

Thank you for your consideration.

If one of the referees has suggested that your manuscript should undergo extensive English revisions, please address this issue during revision. We propose that you use one of the editing services listed at https://www.mdpi.com/authors/english or have your manuscript checked by a native English-speaking colleague.

Thanks for the positive comments of you and all reviewers for our manuscript. The English language of the manuscript has been well modified by Charlesworth Author Services (https://www.cwauthors.com.cn/). We then have re-submitted the manuscript.

Thank you for your consideration.

Reviewer 1:

Replace with “harsh condition”

Thanks for your positive comments. As suggested, we have modified the corresponding contents in Abstract Section, namely “Studying the genomic regions controlling maize deep-sowing tolerance would benefit for developing new varieties to resistant in harsh conditions” in Lines 13-15. We then have re-submitted the manuscript.

Thank you for your consideration.

Please mention the climate change/warming in this paragraph.

Thanks for your positive comments. As suggested, we have re-added the corresponding contents “Relative rates of yield increase in maize are slowing and are estimated at about 1.5% per year [1]. Meanwhile, climate change has negatively impacted agricultural production (http://faostat.fao.org/), this will not meet the needs of a rapidly increasing global population; the mean growth rate of crop yield must exceed 2.4% per year to feed an estimated global population of 10 billion by 2050s [2].” in Lines 39-43. We then have re-submitted the manuscript.

Thank you for your consideration.

Explain why did you select these three depths?? Why not 5 cm, 10 cm, and 20 cm?

Thanks for your positive comments. In general, the normal sowing depth of maize seeds is 2-3 cm in the field (Zhao et al. Molecular mechanisms of mesocotyl elongation induced by brassinosteroid in maize under dee-seeding stress by RNA-sequencing, microstructure observation, and physiological metabolism. Genomics 113, 3565-3581). The soil at this seeded layer has the abundant evaporation, low soil water, and scarce precipitation, thereby, the maize seeds are often unable to obtain sufficient water for germination or to ensure normal seedlings growth (Liu et al. Quantitative trait locus analysis for deep-sowing germination ability in the maize IBM Syn10 DH population. Front. Plant Sci. 2017, 8, 813.). Fortunately, “40107” and “P1213733” maize varieties have a strong deep-sowing tolerance in the deep soil layer (> 20 cm), absorbing water from the moist soil, thus effectively avoid drought stress at the early seedling stage (Troyer AF. The location of genes governing long first internode of corn. Genetics 1997, 145, 1149–1154. Zhao et al. The combination of Conventional QTL analysis, bulked-segregant analysis, and RNA-sequencing provide new genetic insights into maize mesocotyl elongation under multiple deep-seeding environments. Int. J. Mol. Sci. 2022, 23, 4223.).

In addition, to analyze the molecular mechanism and identify QTLs for deep-sowing tolerance in maize, Liu et al. (Quantitative trait locus analysis for deep-sowing germination ability in the maize IBM Syn10 DH population. Front. Plant Sci. 2017, 8, 813.) used 280 IBM Syn10 DH population to measure six deep-sowing tolerant traits under two sowing depths (2 cm and 12.5 cm), and then identified 65 QTLs. Zhang et al. (Mapping QTL controlling maize deep-seeding tolerance-related traits and confirmation of major QTL for mesocotyl length. Theor. Appl. Genet. 2012, 124, 223–232.) developed 221 F2:3 families population to measure four deep-sowing tolerant traits under two sowing depths (10 cm and 15 cm), and then mapped 25 QTLs. Zhao et al. (The combination of Conventional QTL analysis, bulked-segregant analysis, and RNA-sequencing provide new genetic insights into maize mesocotyl elongation under multiple deep-seeding environments. Int. J. Mol. Sci. 2022, 23, 4223.) established 346 F2:3 families population to evaluate one deep-sowing tolerant trait under three sowing depths (3 cm, 15 cm, and 20 cm), and then found 7 QTLs.

Therefore, “Based on above considerations, the objectives of our study were to: (1) identify QTLs information responsible for six deep-sowing tolerant traits in 346 F2:3 population families derived from a W64A × K12 cross at three sowing depths (3 cm, 15 cm, and 20 cm)” in Lines 88-91.

Thank you for your consideration.

What do you think about water consumption of maize in deep sowing while water is insufficient in the glob?

Thanks for your positive comments. It is reported that the soil moisture of deeper soil layer accounts for approximately 15% in (semi)-arid areas of China (Zhao et al. Transcriptomic and metabolic profiling reveals a lignin metabolism network involved in mesocotyl elongation during maize seed germination. Plants 2022, 11, 1034). Therefore, in these regions, it is necessary to sow seeds deep to reach the moist soil required for better seedling emergence, and seeds will absorb water from deep soil layer and emerge normally (Zhao et al. 24-epibrassinolide mediated interaction among antioxidant defense, lignin metabolism, and phytohhormones signaling promoted better cooperative elongation of maize mesocotyl and coleoptile under deep-seeding stress. Russ. J. Plant Physiol. 2021, 68, 1194-1207.).

In addition, it is well known that “development of maize photosynthetic apparatus remains undisturbed through protection of the shoot apex from cold exposure at deeper soil level [6]. A deeper soil environment may also allow maize to evade negative effects of chemical residues in shallow soil and damage by wild animals [7], thereby stabilizing the primary form of seedlings [8]. However, due to the inherent characteristics, most existing varieties have weak germination potential in deep soil environments [9]. Thus, improving deep-sowing tolerance would be a key strategy to maintain normal growth and high yield of maize under cold or drought conditions.” in Lines 49-56.

Thank you for your consideration.

Strange to me at first Results and then Materials and Methods!!!

Thanks for your positive comments. According to the submission guide of the International Journal of Molecular Sciences (https://www.mdpi.com/journal/ijms), we wrote the manuscript and typeset the manuscript. The layout is exactly consistent with the published articles in this Journal (https://www.mdpi.com/journal/ijms).

Thank you for your consideration.

This is Material.

Thanks for your positive comments. Based on the submission guide of the International Journal of Molecular Sciences (https://www.mdpi.com/journal/ijms), we wrote the manuscript and typeset the manuscript. The layout is exactly consistent with the published articles in this Journal (https://www.mdpi.com/journal/ijms). Thus Section 2 was arranged the Results Section rather than Materials Section.

Thank you for your consideration.

Did you check the data for normal distribution?

Thanks for your positive comments. Yes, we have checked the data for normal distribution in Table 1 (including skewness and kurtosis values). At the same time, in Results Section, we have described in detail the corresponding contents, namely “The values of all six deep-sowing tolerant traits were distributed normally in the F2:3 population at all three sowing depths, with absolute values of skewness and kurtosis of < 1.0, except for COLL and MESL/COLL at 20 cm depth (skewness = 1.087 and 0.913, kurtosis = 0.934 and 1.017, respectively) (Table 1).” in Lines 130-133.

Thank you for your consideration.

Please don’t repeat the results in the discussion section.

Thanks for your positive comments. As suggested, we have deleted the corresponding contents. We then have re-submitted the manuscript.

Thank you for your consideration.

Moderate English changes required.

Thanks for your positive comments. As suggested, we have carefully checked the English language of the manuscript and have then perfected the English language. We then have re-submitted the manuscript.

Thank you for your consideration.

Does the introduction provide sufficient backaround and include all relevant references? Yes

Thanks for your positive comments.

Thank you for your consideration.

Are all the cited references relevant to the research? Can be improved.

Thanks for your positive comments. As suggested, we have checked and modified the cited references (Lines 771-927). We then have re-submitted the manuscript.

Thank you for your consideration.

Is the research design appropriate? Yes

Thanks for your positive comments.

Thank you for your consideration.

Are the methods adequately described? Yes

Thanks for your positive comments.

Thank you for your consideration.

Are the results clearly presented? Yes

Thanks for your positive comments.

Thank you for your consideration.

Are the conclusions supported by the results? Yes

Thanks for your positive comments.

Thank you for your consideration.

Reviewer 2:

The Integrated Mapping, Meta-Analysis, and RNA-Sequencing Reveal Candidate Genes for Maize Deep-Sowing Tolerance presents scientific merit with robust results, with significant advances in the knowledge area. Therefore my recommendation is to be filed for publication in the current format.

Thanks for your positive comments and recognition of our work.

Thank you for your consideration.

I would like to sign my review report.

Thanks for your positive comments.

Thank you for your consideration.

I am not qualified to assess the quality of English in this paper.

Thanks for your positive comments. We have carefully checked the English language of the manuscript and have then perfected the English language again. We then have re-submitted the manuscript.

Thank you for your consideration.

Does the introduction provide sufficient backaround and include all relevant references? Yes

Thanks for your positive comments.

Thank you for your consideration.

Are all the cited references relevant to the research? Yes

Thanks for your positive comments.

Thank you for your consideration.

Is the research design appropriate? Yes

Thanks for your positive comments.

Thank you for your consideration.

Are the methods adequately described? Yes

Thanks for your positive comments.

Thank you for your consideration.

Are the results clearly presented? Yes

Thanks for your positive comments.

Thank you for your consideration.

Reviewer 3:

The author of this article used 346 F2:3 maize populations and 253 SSRs to identify 33 quantitative QTLs related to Deep-Sowing Tolerance using composite interval mapping (CIM) under three sowing depths. The validation of the genes using qPCR and obtained satisfactory results. The article was well-designed with clear thinking, and it can be seen that the author team has put in a lot of work. However, some issues need to be addressed and explained by the author.

Thanks for your positive comments. As suggested, we have improved and modified the manuscript. We then have re-submitted the manuscript. Moreover, we also answered all your questions.

Thank you for your consideration.

  1. In the introduction, the author mentioned several traits related to Deep-Sowing Tolerance, but did not elaborate on why.

Thanks for your positive comments. “Different crop species have evolved diverse organs to adapt to deep-sowing stressors, including elongated mesocotyl in rice (Oryza sativa L.) [10] and elongated coleoptile in wheat (Triticum aestivum L.) [11] and barley (Hordeum vulgare L.) [12]. These structures help emerge the shoot tips and the first internode from the embryo to the soil surface; the co-elongated mesocotyl and coleoptile in maize send the buds to the soil surface [13,14], where natural light represses the continued elongation of mesocotyl and coleoptile and induces leaf expansion [7,15].” in Lines 58-63.

In addition, a large number of studies have revealed and confirmed the relationship between deep-sowing tolerance and phenotypic traits including emergence rate (RAT), mesocotyl length (MESL), coleoptile length (COLL), total length of mesocotyl and coleoptile (MESL+COLL), length ratio of mesocotyl to coleoptile (MESL/COLL), mesocotyl coarse (MESC), coleoptile coarse (COLC), mesocotyl fresh weight (MESW), coleoptile fresh weight (COLW), seedling length (SDL), root length (RL), and root weight (RW) under different sowing depths (Zhao et al. New insights into light spectral quality inhibits the plasticity elongation of maize mesocotyl and coleoptile during seed germination. Front. Plant Sci. 2023, 14, 1152399; Zhong et al. Heterosis and genetic effects analysis of deep-seeding traits in maize under different sowing environments. J. Nuclear Agric. Sci. 2021, 35, 556-566; Liu et al. Quantitative trait locus analysis for deep-sowing germination ability in the maize IBM Syn10 DH population. Front. Plant Sci. 2017, 8, 813; Peng et al. Deep-sowing tolerance and genetic diversity of maize inbred lines. Acta Prataculturae Sin. 2016, 25, 73-86; Peng et al. Deep-sowing tolerant characteristics in maize inbred lines. Agric. Res. The Arid Areas 2014, 32, 25-33.), resulting in these traits are important deep-sowing tolerant traits, which can be good indicators for screening of deep-sowing tolerant genotypes of maize. Hence, in recent years, some QTL mapping for some above deep-sowing tolerant traits including RAT, MESL, COLL, SDL, and plumule length (PL) at three sowing depths of 2 cm, 3 cm, 12.5 cm, 15cm, and 20 cm (Zhang et al. Mapping QTL controlling maize deep-seeding tolerance-related traits and confirmation of major QTL for mesocotyl length. Theor. Appl. Genet. 2012, 124, 223–232; Han et al. QTL analysis of deep-sowing tolerance during seed germination in the maize IBM Syn4 RIL population. Plant Breed. 2020, 139, 1125–1134; Liu et al. Quantitative trait locus analysis for deep-sowing germination ability in the maize IBM Syn10 DH population. Front. Plant Sci. 2017, 8, 813; Zhao et al. The combination of Conventional QTL analysis, bulked-segregant analysis, and RNA-sequencing provide new genetic insights into maize mesocotyl elongation under multiple deep-seeding environments. Int. J. Mol. Sci. 2022, 23, 4223.).

Based on the above considerations, in the Introduction Section of this study, we mentioned these deep-sowing tolerant traits. Therefore, “the objectives of our study were to: (1) identify QTLs information responsible for six deep-sowing tolerant traits in 346 F2:3 population families derived from a W64A × K12 cross at three sowing depths (3 cm, 15 cm, and 20 cm), (2) incorporate previously published data in a meta-analysis to integrate genetic maps and identify corresponding MQTLs involving in deep-sowing tolerance in maize, (3) mine candidate genes in corresponding MQTLs and major QTLs regions by combining RNA-Seq and quantitative real-time PCR (qRT-PCR) in mesocotyl and coleoptile of both parents (W64A and K12) at 3 cm and 20 cm sowing depths environments, and (4) further explore the interconnected networks of candidate genes and interpret the biological processes and pathways controlling maize deep-sowing tolerance. These findings will provide valuable information by revealing molecular basis of deep-sowing tolerance in maize and annotation of associated genes that will lay a foundation for further functional analysis of genes regulating deep-sowing tolerance in maize and the discovery of alleles to be used in developing deep-sowing resistant varieties.” in Lines 88-101.

Thank you for your consideration.

  1. In the materials and methods section, the author did not mention the number of replicates for each Deep-Sowing treatment and the number of individuals per replicate, which is important information for locating a quantitative trait.

Thanks for your positive comments. In the Materials and Methods Section, we had specified the number of sowing environments (n, n = 3), the number of replications (r, r = 10), and the number of measured seedlings (No. = 10).namely “Ten days after germination, RAT, MESL, COLL, MESL+COLL, MESL/COLL, and SDL of ten seedlings were measured at each sowing environments.” in Lines 587-589, and The broad-sense heritability ( ) and genotype × environment interaction heritability ( ) values of above six traits in F2:3 population were calculate as follows [66,67]: HB2 = sg2/(sg2 + sge2/n + se2/nr), HGE2 = (sg2/n)/(sg2 + sge2/n + se2/nr). Where sg2 was genotypic variance, se2 was environmental variance, se2 was error variance, sge2 was variance of genotype × environment interaction, n (n = 3) was the number of sowing environments, and r (r = 10) was the number of replications.” in Lines 594-599. We then have re-submitted the manuscript.

Thank you for your consideration.

  1. In the results section 1, it is recommended to use distribution plots instead of Table 1 to more intuitively understand the distribution of each trait in the population.

Thanks for your positive comments. In the results Section 1, Table 1 showed in detail the Mean±SD (Standard deviation) of six deep-sowing tolerant traits (including emergence rate (RAT), mesocotyl length (MESL), coleoptile length (COLL), total length of mesocotyl and coleoptile (MESL+COLL), length ratio of mesocotyl to coleoptile (MESL/COLL), and seedling length (SDL)) in female parent (W64A), male parent (K12), F1 hybrid (W64A × K12), and F2:3 populations under three sowing depths environments (including 3 cm, 15 cm, and 20 cm), and the range, coefficient of variation (CV), Skewness, and Kurtosis of these six deep-sowing tolerant traits in F2:3 population under each sowing depth environments. There is no doubt that distribution plots will not show these information in Table 1 accurately. Moreover, these information in Table 1 can also avoid taking up too much space because of the figures.

Using the same table, Zhang et al. (Mapping QTL controlling maize deep-seeding tolerance-related traits and confirmation of major QTL for mesocotyl length. Theor. Appl. Genet. 2012, 124, 223–232) also showed in detail the Mean of four deep-sowing tolerant traits (including RAT, MESL, COLL, SDL) in female parent (X178), male parent (3681-4), and F2:3 families under two sowing depths environments (including 10 cm and 20 cm), and the range, Skewness, Kurtosis, and heritability of these four deep-sowing tolerant traits in F2:3 population under each sowing depth environments.

Thank you for your consideration.

  1. In the results section 2, regarding the heritability, it is inappropriate to treat the three Deep-Sowing treatments as environment factor to calculate genotype × environment interaction heritability. The appropriate approach is to conduct Deep-Sowing on the population under different planting environments. If the author intends to compare the differences in the difference between Deep-Sowing treatments, a t-tests should be used instead of comparing heritability differences.

Thanks for your positive comments. Previously, Zhao et al. (The combination of Conventional QTL analysis, bulked-segregant analysis, and RNA-sequencing provide new genetic insights into maize mesocotyl elongation under multiple deep-seeding environments. Int. J. Mol. Sci. 2022, 23, 4223.) used the measured methods of broad-sense heritability ( ) and genotype × environment interaction heritability ( ) values, i.e., “Values of  and  for MES among all conditions were estimated by Equations (1) and (2), as follows [39]: HB2 = sg2/(sg2 + sge2/n + se2/nr), (1). HGE2 = (sg2/n)/(sg2 + sge2/n + se2/nr), (2). Where, sg2 was the genotypic variance; se2 was the environmental variance; se2 represented the error variance, sge2 was the variance of genotype × environment interaction; n was the number of sowing depth environments (n = 3); and r was the number of replications ( r = 10).” to estimate the  and  values of mesocotyl length (MESL) in F2:3 families under three sowing depth environments of 3 cm, 15 cm, and 20 cm. Similarly, Khan et al. (Analysis of QTL-allele system conferring drought tolerance at seedling stage in a nested association mapping population of soybean [Glycine max (L.) Merr.] using a novel GWAS procedure. Planta 2018, 248, 947-962.) used this method to estimate the   and  values of root length and shoot length in a nested association population (NAM) under polyethylene-glycol (PEG) and non-PEG treatments.

Using the same method, in this study, we also estimated the  and  values of six deep-sowing tolerant traits including emergence rate (RAT), mesocotyl length (MESL), coleoptile length (COLL), total length of mesocotyl and coleoptile (MESL+COLL), length ratio of mesocotyl to coleoptile (MESL/COLL), and seedling length (SDL) in a 346 F2:3 families population under three sowing depths environments of 3 cm, 15 cm, and 20 cm.

Thank you for your consideration.

  1. In the results section, regarding the relationship between various traits, (due to the vague description in the MS), it seems that the author analyzed all the data from three Deep-Sowing treatments together. This is inappropriate. For example, the Deep-Sowing will significantly stimulate the elongation of MESL and COLL and reduce RAT. This may lead to the situation where, compared with SW15, the MESL and COLL of individuals under SW20 may be longer, but the RAT is lower. Therefore, if all the phenotypic data of the three treatments are analyzed together, obvious errors would occur. Like Figure 2-C, the negative correlation between MESL/COLL and COLL should be obvious, but it is shown as a positive correlation in the Figure, and the same phenomenon exists in Figure 2-b. At the same time, I do not recommend using PCA to analyze the relationship between traits because the bias would arise during dimension reduction process.

Thanks for your positive comments. Yes, I totally agreed with you, i.e., the deep-sowing will significantly stimulate the elongation of mesocotyl length (MESL) and coleoptile length (COLL) and reduce emergence rate (RAT) in maize. This may lead to the situation where, compared with 3 cm sowing depth environment (SW3), the MESL and COLL of individuals under 15 cm and 20 cm deep-sowing environments (SW15 and SW20) may be longer, but the RAT is lower. Therefore, the elongation of mesocotyl and coleoptile (i.e., MESL and COLL) in maize was consistent at 15 cm and 20 cm sowing depths.

In these regards, in this study, we performed the Pairwise Pearson correlations among six deep-sowing tolerant traits at 15 cm and 20 cm deep-sowing environments, namely ““Pairwise Pearson correlations showed that six deep-sowing tolerant traits produced complementary information at 15 cm and 20 cm deep-sowing environments, and each trait had a significantly positive or negative correlation (p < 0.05) with 2–5 other traits (Figure 2C).” in Lines 179-182. In addition, we also performed the principal component analysis (PCA) among six deep-sowing tolerant traits at 3 cm, 15 cm, and 20 cm sowing depths environments, namely “PCA analysis showed that first two principal components (PCs, PC1 and PC2) accounted for 94.1% of the total variance under three environments, of which the eigenvalues were larger than 1.0 (Figure 2A). We therefore speculated that the two PCs linear combinations of different traits were based on their variable loadings. Specifically, MESL, COLL, MESL+COLL, and MESL/COLL were the primary traits in PC1, which accounted for 52.7% of the total variance and represented the developmental characteristics of mesocotyl and coleoptile in maize; RAT and SDL were the primary traits in PC2, which accounted for 41.4% of the total variance and indicated the emergence and growth of maize seedlings (Figure 2B).” in Lines 168-179. Therefore, this results confirmed that RAT and SDL indicated the emergence and growth of maize seedlings, and other four traits indicated the developmental characteristics of mesocotyl and coleoptile in maize.

 Thank you for your consideration.

  1. The author has been describing six related traits, but from Table 3 onwards, Deep-Sowing Tolerance-related traits are changed to seven, without any explanation.

Thanks for your positive comments. Yes, In this study, we described the phenotypic values of six deep-sowing tolerant traits including emergence rate (RAT), mesocotyl length (MESL), coleoptile length (COLL), total length of mesocotyl and coleoptile (MESL+COLL), length ratio of mesocotyl to coleoptile(MESL/COLL), and seedling length (SDL) in a 346 F2:3 families under three sowing environments and then identified QTLs for these traits. These results were showed in Table 1, Figure 3, and Table S1.

Moreover, we knew that these deep-sowing tolerant traits also included other traits, for example, plumule length (PL). Therefore, we also collected the original QTLs for RAT, MESL, COLL, MESL+COLL, MESL/COLL, PL, and SDL in maize from present and previous studies to detect meta-QTLs and candidate genes for deep-sowing tolerance.  Table 3 summarized the relevant original QTLs information from present and previous studies, so there were seven deep-sowing tolerant traits.

Thank you for your consideration.

  1.  In Figure 4-A, the QTL frequency distribution is not related to the main theme of the article and can be removed or placed in the supplementary.

Thanks for your positive comments. Figure 4A showed the QTL frequency distribution, which can provide some important information including the distribution of QTLs for each trait or all deep-sowing tolerant traits on ten chromosomes. Therefore, these information can guide us to purposefully increase the molecular markers of corresponding chromosomes and fine map several major QTLs for target traits.

Similarly, Zhao et al. (QTL mapping for six ear leaf architecture traits under water-stressed and well-watered conditions in maize (Zea mays L.). Plant Breed. 2018, 137, 60-72.) used this strategy to find the corresponding important chromosomes regulating six leaf architecture traits in maize, then they purposefully selected molecular markers to construct genetic map and perform QTL mapping for six ear leaf architecture traits under water-stressed and well-watered conditions in maize.

Thank you for your consideration.

  1. In Figures 4-B and C, it is not recommended to use distribution plots to represent QTL LOD and PVE.

Thanks for your positive comments. It is well known that “meta-QTL (MQTL) analysis has used to estimate consensus QTLs linked to genetic factors underlying deep-sowing tolerant traits among multiple independent QTL mapping experiments [22–24].” in Lines 80-82. When performing meta-QTL analysis, original QTLs information should be collected, namely “These QTLs information mainly included original chromosomal position, LOD scores, confidence interval (CI), and proportion of PVE.” in Lines 630-631. Similarly, Zhao et al. (Identification of QTLs and meta-QTLs for seven agronomic traits in multiple maize populations under well-watered and water-stressed conditions. Crop Sci. 2018, 58, 507-520) and Zhao et al. (Mapping QTLs and meta-QTLs for two inflorescence architecture traits in multiple maize populations under different watering environments. Mol. Breed. 2017, 37, 91.) to analyze meta-QTLs for seven agronomic traits or two inflorescence architecture traits in maize, they first collected original QTLs including original chromosomal position, LOD scores, confidence interval (CI), and proportion of PVE.

In summary, the distribution plots of QTL LOD and PVE are important to meta-QTL analysis.

Thank you for your consideration.

  1. The 33 identified QTLs are too many for only 253 SSRs, and many loci with low PVE should be removed, and the number of QTLs should be kept to less than five.

Thanks for your positive comments. Previously, Zhang et al. (Mapping QTL controlling maize deep-seeding tolerance-related traits and confirmation of major QTL for mesocotyl length. Theor. Appl. Genet. 2012, 124, 223–232.) used 158 SSRs and 21 IDP amplified polymorphic bands to identified 25 QTLs for four deep-sowing tolerant traits in a 221 F2:3 families population under two sowing depths environments, which explained 3.81-17.95% of phenotypic variance.

Similarly, in this study, “we detected QTLs using composite interval mapping (CIM) approach and the threshold logarithm of odds (LOD) score of 3.00 (p < 0.05) in Windows QTL Cartographer software v.2.5 (https://brcwebportal.cos.ncsu.edu/qtlcart/WQTLCart.htm; accessed on 21 Jul. 2022). We identified 33 QTLs for six deep-sowing tolerant traits (four for RAT, seven for MESL, five for COLL, eight for MESL+COLL, three for MESL/COLL, and six for SDL) in the F2:3 population at three sowing depths, which were distributed over the ten chromosomes; phenotypic variation explained (PVE) within each sowing environment by these QTLs were 2.89% (qMESL–Ch.1–1 at 3 cm environment) – 14.17% (qSDL–Ch.7–1 at 20 cm environment)(Figure 3; Table S1).” in Lines 227-235.

Thank you for your consideration.

I would like to sign my review report.

Thanks for your positive comments.

Thank you for your consideration.

I am not qualified to assess the quality of English in this paper.

Thanks for your positive comments. We have carefully checked the English language of the manuscript and have then perfected the English language again. We then have re-submitted the manuscript.

Thank you for your consideration.

Does the introduction provide sufficient backaround and include all relevant references? Can be improved

Thanks for your positive comments. As suggested, we have further improved the Introduction Section in Lines 35-101. We then have re-submitted the manuscript.

Thank you for your consideration.

Are all the cited references relevant to the research? Yes

Thanks for your positive comments.

Thank you for your consideration.

Is the research design appropriate? Must be improved

Thanks for your positive comments. As suggested, we have improved the experiment design in Lines 88-101. We then have re-submitted the manuscript.

Thank you for your consideration.

Are the methods adequately described? Must be improved

Thanks for your positive comments. As suggested, we have further improved the Methods Section in Lines 569-671. We then have re-submitted the manuscript.

Thank you for your consideration.

Are the results clearly presented? Must be improved

Thanks for your positive comments. As suggested, we have further improved the Results Section in Lines 102-336. We then have re-submitted the manuscript.

Thank you for your consideration.

Are the conclusions supported by the results? Can be improved

Thanks for your positive comments. As suggested, we have further improved the Conclusions Section in Lines 672-692. We then have re-submitted the manuscript.

Thank you for your consideration.

Sincerely,

Xiaoqiang Zhao professor

State Key Laboratory of Aridland Crop Science, Gansu Agricultural University

E-mail: zhaoxq3324@163.com

Reviewer 2 Report

The Integrated Mapping Manuscript, Meta-Aalysis, and RNA-SEQUENCING Reveal Candidate Genes for Maize Deep-Sowing Tolerance presents scientific merit with robust results, with significant advances in the knowledge area. Therefore my recommendation is to be filed for publication in the current format.

Author Response

(The authors gave the same response as above.)

Reviewer 3 Report

The author of this article used 346 F2:3 maize populations and 253 SSRs to identify 33 quantitative QTLs related to Deep-Sowing Tolerance using composite interval mapping (CIM) under three sowing depths. The validation of the genes using qPCR and obtained satisfactory results. The article was well-designed with clear thinking, and it can be seen that the author team has put in a lot of work. However, some issues need to be addressed and explained by the author.

1.    In the introduction, the author mentioned several traits related to Deep-Sowing Tolerance, but did not elaborate on why.

2.    In the materials and methods section, the author did not mention the number of replicates for each Deep-Sowing treatment and the number of individuals per replicate, which is important information for locating a quantitative trait.

3.    In the results section 1, it is recommended to use distribution plots instead of Table 1 to more intuitively understand the distribution of each trait in the population.

4.    In the results section 2, regarding the heritability, it is inappropriate to treat the three Deep-Sowing treatments as environment factor to calculate genotype × environment interaction heritability. The appropriate approach is to conduct Deep-Sowing on the population under different planting environments. If the author intends to compare the differences in the difference between Deep-Sowing treatments, a t-tests should be used instead of comparing heritability differences.

5.    In the results section, regarding the relationship between various traits, (due to the vague description in the MS), it seems that the author analyzed all the data from three Deep-Sowing treatments together. This is inappropriate. For example, the Deep-Sowing will significantly stimulate the elongation of MESL and COLL and reduce RAT. This may lead to the situation where, compared with SW15, the MESL and COLL of individuals under SW20 may be longer, but the RAT is lower. Therefore, if all the phenotypic data of the three treatments are analyzed together, obvious errors would occur. Like Figure 2-C, the negative correlation between MESL/COLL and COLL should be obvious, but it is shown as a positive correlation in the Figure, and the same phenomenon exists in Figure 2-b. At the same time, I do not recommend using PCA to analyze the relationship between traits because the bias would arise during dimension reduction process.

6.    The author has been describing six related traits, but from Table 3 onwards, Deep-Sowing Tolerance-related traits are changed to seven, without any explanation.

7.    In Figure 4-A, the QTL frequency distribution is not related to the main theme of the article and can be removed or placed in the supplementary.

8.    In Figures 4-B and C, it is not recommended to use distribution plots to represent QTL LOD and PVE.

9.    The 33 identified QTLs are too many for only 253 SSRs, and many loci with low PVE should be removed, and the number of QTLs should be kept to less than five.

Author Response

(The authors gave the same response as above.)
